# Circumpolar spread of avian influenza H5N1 to southern Indian Ocean islands

Augustin Clessin [1], François-Xavier Briand[2], Jérémy Tornos [1], Mathilde Lejeune[1], Camille De Pasquale [1], Romain Fischer[1], Florent Souchaud [2], Edouard Hirchaud[2], Samuel L. Hong [3], Tristan Bralet[1,4], Christophe Guinet [5], Clive R. McMahon [6], Béatrice Grasland[2], Guy Baele [3] & Thierry Boulinier [1] ✉

Since 2020, the outbreak of high pathogenicity avian influenza (HPAI) H5N1 virus clade 2.3.4.4b has turned into the largest documented panzootic [1,3]. Here, we describe its arrival into the Indian Ocean sub-Antarctic archipelagos of Crozet and Kerguelen, where we first detected the virus in October 2024 in dead southern elephant seals. While the panzootic is ongoing, it has already caused unprecedented mortalities of marine mammals and seabirds. We collected brain swabs from seal and seabird carcasses and obtained 25 novel HPAI H5N1 2.3.4.4b sequences. Using phylogeographic analyses, we show that there have been independent introductions of the virus to Crozet and Kerguelen islands, most likely from the distant South Georgia islands in the Southern Atlantic, and not from the more nearby coasts of South Africa. Our results point to a year-long gap in genomic surveillance in the sub-Antarctic region. Locally, our analyses show that the virus is transmitted between different species. Our serological analyses show that some southern elephant seal had mounted an anti-H5 antibody response. Through its circumpolar spread to the Indian Ocean, HPAI H5N1 2.3.4.4b moves closer to Australia, which remains free from infections with this strain, and represents a major threat to the sub-Antarctic wildlife.

The ongoing outbreak of clade 2.3.4.4b of high pathogenicity avian influenza (HPAI) H5N1 virus has been unique not only in the large number of host species affected, with a wide range of both birds and mammals reported dead[1–3], but also in the scale of the panzootic, with all continents except Oceania affected[4]. Originating in Asia in 1996[5,6], the A/Goose/Guangdong/1/1996 lineage HPAIV spread across the world in different waves, recombined and diversified into different clades of various H5Nx viruses. Clade 2.3.4.4b started to become a worldwide preoccupation in 2014, and in the large outbreak of 2016-

2017, HPAI H5N8 viruses of clade 2.3.4.4b reached South Africa[7,8]. Since 2021, HPAI H5N1 viruses of clade 2.3.4.4b severely impacted wild bird species and poultry across Europe[9–11], southern Africa[12–14], North America[15], and South America where it resulted in die-offs of tens of thousands of seabirds[16] and marine mammals[17,18]. At the time of writing, HPAIV H5N1 clade 2.3.4.4b has not been detected in any birds or other animals in Australia and the southern Pacific region[19]. While there were no incursions into the Antarctic region during the summer season 2022/2023[20], HPAIV H5N1 clade 2.3.4.4b was detected for the first time

[1]Centre d'Ecologie Fonctionnelle et Evolutive (CEFE), CNRS, Université Montpellier, EPHE, IRD, Montpellier, France. [2]ANSES, Ploufragan-Plouzané-Niort Laboratory, Laboratoire National de Référence, Unité de virologie, Immunologie, Parasitologie, Aviaires et Cunicoles, Ploufragan, France. [3]Department of Microbiology, Immunology and Transplantation, Rega Institute, KU Leuven, Leuven, Belgium. [4]ANSES, UZB, Maisons-Alfort, France. [5]Centre d'Etudes Biologiques de Chizé (CEBC), UMR 7372, CNRS-La Rochelle Université, Villiers-en-Bois, France. [6]IMOS Animal Tagging, Sydney Institute of Marine Science, Mosman, New South Wales, Australia. ✉e-mail: thierry.boulinier@cefe.cnrs.fr

in the Antarctic continent and the surrounding sub-Antarctic islands in October, 2023[4], when it spread from the tip of South America to the Falkland and South Georgia islands. On route down the coast of South America, the virus had a devastating effect on seal and seabird populations[16,21], hence the scientific community was extremely concerned about the impact the virus could have on the dense seabird and pinniped communities of the sub-Antarctic and Antarctic regions. The situation on South Georgia Island turned quickly into a mass mortality event of pinnipeds and seabirds. The mortality levels were especially worrying for two species, the snowy albatross (*Diomedea exulans*)[22] and the southern elephant seals (*Mirounga leonina*)[4]. Moreover, scavenging species, like the brown skua (*Stercorarius antarcticus*) and the giant petrels (*Macronectes* spp), have been suspected to play a key role in virus spread[20].

The sub-Antarctic vertebrate communities are characterised by pinnipeds and seabirds that forage at sea and breed in dense and often mixed-species colonies on the sparse islands and archipelagos that span the region[23]. This results in an annual pulse of high numbers of concentrated animal congregations to breed and moult. For instance, the French Southern Lands (Crozet, Kerguelen, Saint-Paul and Amsterdam islands) are home to 50 million seabirds of up to 47 species. They aggregate in large numbers for breeding on Crozet and Kerguelen islands each year between October and March, along with hundreds of thousands of marine mammals[24]. In addition to being at risk of massive die-offs because they are densely packed during breeding, several of the seabird populations are especially threatened because they are only present at a few locations, and their populations are already under pressure from other threats[25]. Seabirds and seals make extensive movements at sea within and outside the breeding season, and it is critical to track the potential spread of the virus over vast areas of oceanic water and identify its mechanisms of transmission within and among wild host communities[26].

The Crozet and Kerguelen archipelagos are both located in the Southern Indian Ocean and are among the most isolated islands of the world, being respectively 2300 and 3800 km from the nearest continents. While much has been learnt about the at-sea movements of seabirds and seals from and to the archipelagos[27], it is unclear how these movements may facilitate the spread of disease agents like H5N1, given the limited information on infectious period length, mortality rates and pathogeny in key potential hosts. Consequently, and despite growing concerns of the scientific community given the extent of the HPAI epizootic in the Atlantic sub-Antarctic region, the eventual arrival of the virus to the sub-Antarctic Indian region remained feared and uncertain during the entire 2023-2024 austral summer[20]. In October 2024, we recorded abnormally high numbers of deaths in southern elephant seal colonies on Possession Island, in the Crozet archipelago, Southern Indian Ocean (Fig. 1). Later, we recorded deaths of several seabird species in lower numbers. The first abnormal mortality of southern elephant seals was detected almost a month later on Kerguelen Island. We here describe the first confirmed HPAI outbreaks on the Crozet and Kerguelen archipelagos and report the results of phylogeographic analyses of the virus sequences we obtained. This provides us with key insights into the origin and number of introductions and on the similarities of viral isolates from mammal and seabird carcasses. Initial serological investigations suggest that a number of southern elephant seals mounted an antibody response.

## Results
### Descriptive epidemiology

The abnormal die-offs of southern elephant seal pups on Possession Island, Crozet archipelago, were detected by one of us (CDP) in the third week of October 2024. On the following days, mortalities also included several adult elephant seals, king penguins, brown skuas, snowy albatrosses, gentoo penguins (*Pygoscelis papua)* and a Cape

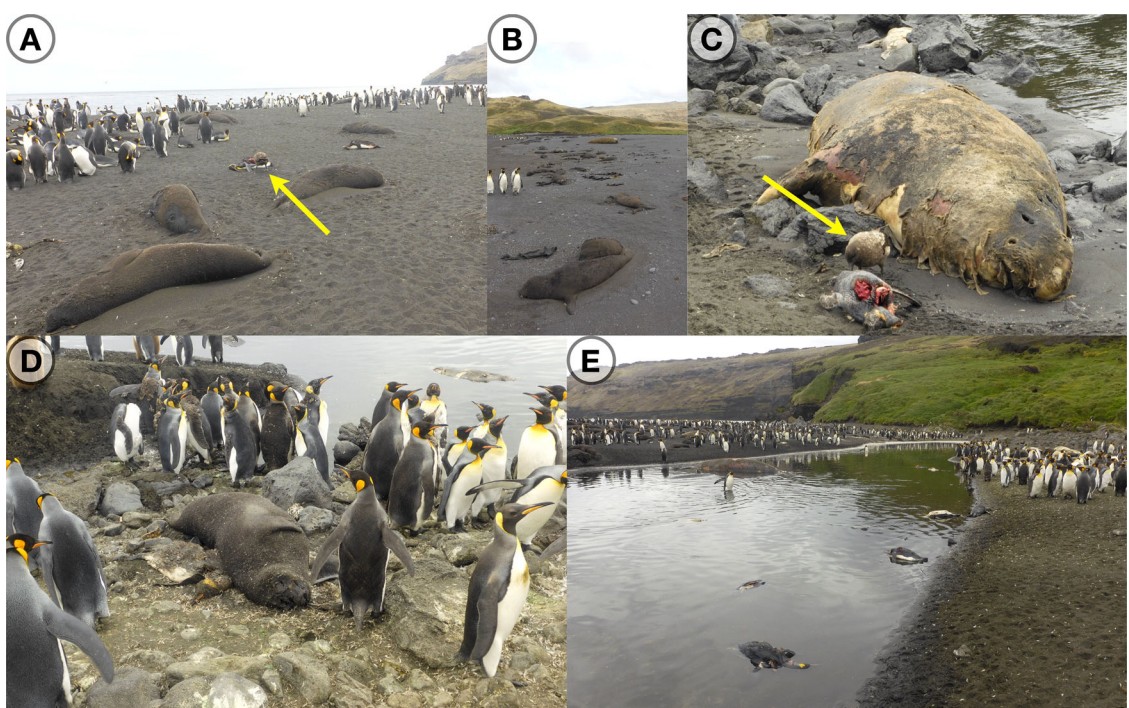

**Fig. 1 | Carcasses of southern elephant seals and king penguins on Petite Manchotière, Possession Island, Crozet archipelago. A** Brown skua scavenging on a penguin carcass (yellow arrow). **B** Aggregation of dead elephant seal pups. **C** Brown skua scavenging on a penguin carcass (yellow arrow) next to a dead adult elephant seal. **D** Dead elephant seals and king penguins in very close proximity to high densities of king penguins. **E** Carcasses of king penguins at the river mouth at the edge of a large breeding colony. We refer to Extended Data Fig. 1 for more detailed pictures of **A** and **C**, to Extended Data Fig. 2 for a map that shows the location of Crozet and other sub-Antarctic islands, and to Fig. 2 for detailed maps of our sampling sites on Crozet and Kerguelen islands. Picture credit: Jérémy Tornos/ Mathilde Lejeune, CNRS/IPEV.

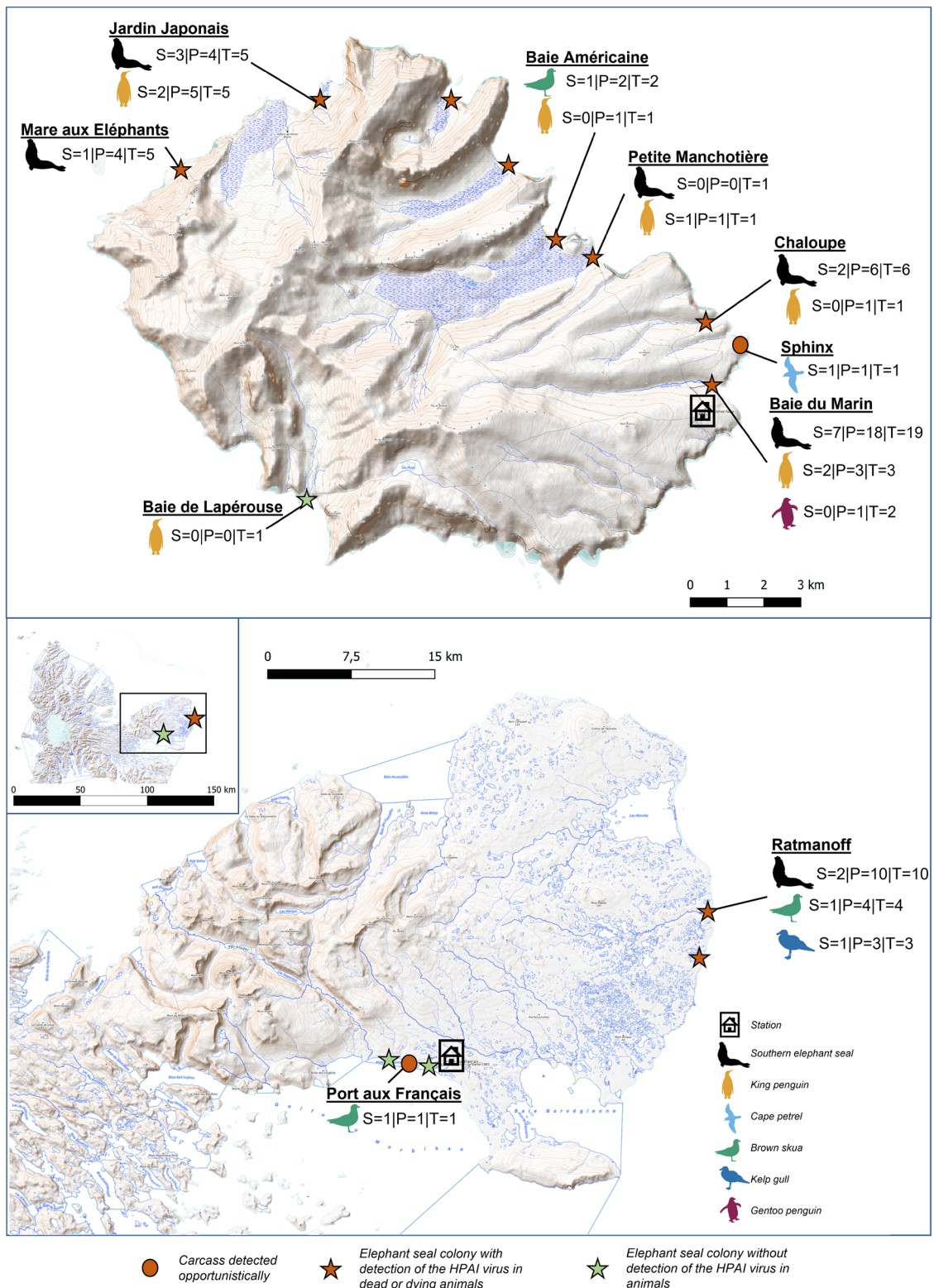

**Fig. 2 | The Crozet (Possession Island) and Kerguelen (Courbet peninsula) islands.** We provide details of our sampling sites (S: number of samples sequenced; P: number of qRT-PCR-positive samples; T: total number of samples analysed) and species. We conducted sampling between October 20th and November 24th, 2024, on Crozet islands, and on the 1st of December 2024 on Kerguelen islands (details in Extended Table 1). Map data copyright OpenStreetMap contributors, licensed under the Open Database License (ODbL) 1.0. Map style copyright OpenTopoMap, licensed under Creative Commons Attribution-ShareAlike 3.0 (CC BY-SA 3.0).

petrel (*Daption capense*) (Fig. 1). The sites where we detected the initial massive die-off are Petite Manchotière and Baie Américaine beach (Fig. 2), where on October 17th, we counted more than 90 dead southern elephant seal pups over a total of approximately 800. Some of the alive elephant seals showed symptoms, mainly convulsion, conjunctivitis, uveitis and nasal discharge. We saw one adult vomiting. On Baie du Marin, near the base, we detected more than 50 dead elephant seal pups and 2 adults on October 28th, as well as dozens of

dead king penguins. On the two days when we conducted sampling at Jardin Japonais (October 27th and November 7th, 2024), we respectively counted 17 dead pups out of a harem of 21 in total, and only 8 live pups out of a total of more than 270 that we had counted a few weeks earlier. Three weeks later, the death rate had risen considerably at all sites, but the exact proportion is difficult to assess, as the carcasses are rapidly washed away by seawater, buried partly in the sand or eaten by scavengers. Nevertheless, we estimate that more than half of the pups died at most of the sites we visited by mid-November, and mortality was still high at that time. We also found significant numbers of king penguins (over 150 detected on the edges of the colony) dead or showing clinical symptomatic signs at Petite manchotière and Jardin Japonais - lethargy, shivering, eye conjunctivitis and nasal discharge. We conducted a full necropsy of a king penguin in the field and observed remarkable macrological signs of pericardial hemorrhage and broad internal tissue inflammation (Extended Data Fig. 3; histological examination was not available at the time of writing). The number of affected penguins was nevertheless small compared to the total number of alive penguins present - tens of thousands at those three large colonies. Note that mostly moulting adult king penguins were affected. We detected mortalities at many southern elephant seal breeding sites and several seabird breeding sites on Possession Island, including 2 nestlings of snowy albatross (positive with antigenic test against NP avian influenza protein) on November 6th at Petite manchotière. Further intense mortalities of southern elephant seals carried on after those early reports.

On mainland Kerguelen, on November 11th at the Ratmanoff king penguin and southern elephant seal colony, the first dead pup was detected and turned up as avian influenza positive with an antigenic test. On the following days, dozens of dead brown skuas, kelp gulls (*Larus dominicanus*) and southern elephant seal pups were detected on the close perimeter of the large penguin colony. At all sites, scavengers - brown skuas, southern giant petrels (*Macronectes giganteus*), northern giant petrels (*Macronectes halli*) and kelp gulls - were observed foraging on dead carcasses (Fig. 1). We also observed scavenging at sea on floating carcasses by *Procellariiformes* (petrels, shearwaters, albatrosses) and brown skuas. The on-site avian influenza antigenic tests we conducted on brain swabs from dead animals were positive for most sampled individuals or pools of samples, and the presence of HPAI H5N1 virus was confirmed by qRT-PCR conducted at the French National Reference Laboratory.

Across both islands, the virus was detected in 65 samples and sequencing could be conducted on 25 samples from southern elephant seals and seabird species (Extended Data Table 1, Extended Data Fig. 4). A very high proportion of the carcasses tested came out positive for H5N1 by qRT-PCR (overall: 91% [65/71]; 88% [32/36] and 100% [10/10] for the elephants seals on Crozet and Kerguelen islands, respectively, 92% (11/12) for the king penguins on Crozet islands, and 100% (7/7) for the brown skuas over both islands; we also found three kelp gulls and one Cape petrel to be qRT-PCR-positive for HPAI H5N1; see Extended Data Table 1 for details). Among the 21 avian influenza antigenic tests conducted against the avian influenza NP protein, all that came out positive (16) corresponded to carcasses for which at least a positive H5 qRT-PCR sample was found (Extended Data Table 1). Of the four that came out negative and for which a qRT-PCR could be conducted, two came out negative by qRT-PCR, but two - one king and one gentoo penguin - came out positive.

## Phylogenetic analyses

All the viruses detected in our samples are H5N1, clade 2.3.4.4b, genotype B3.2., similarly to what was mostly detected in South America and in south polar areas[17]. Bayesian time-calibrated phylogenetic analysis yielded a mean evolutionary rate of $4.9 \times 10^{-3}$ ($4.6$-$5.2 \times 10^{-3}$ 95% highest posterior density interval; HPD) substitutions per site per year,

in line with the estimate of Uhart et al.[17], who reported a mean avian rate of $5.4 \times 10^{-3}$ substitutions per site per year ($4.9$-$5.9 \times 10^{-3}$ 95% HPD) using a host-specific local clock model[28]. Figure 3A, B show the consensus phylogeny as obtained using the highest independent posterior subtree reconstruction (HIPSTR) approach in TreeAnnotator X[29]. Immediately apparent are the very long branches relating to when HPAI H5N1 clade 2.3.4.4b arrived in the sub-Antarctic region - specifically on South Georgia Island - which is estimated to have taken nearly six months, and which we estimate to have come in from Argentina (but see also Fig. 4 and Extended Data Fig. 5, 6). We estimate subsequent spread eastward in the sub-Antarctic and southward in the Antarctic region - to Antarctica, the Crozet islands and the Kerguelen islands - to have occurred with South Georgia Island as the source (Fig. 3B, C). However, the subsequent dispersal to the (estimated most recent common ancestor in the) Crozet and the Kerguelen islands is estimated to have taken approximately one year from their estimated most recent common ancestor, based on the currently available data. While these islands are geographically very distant, it is remarkable that such long-distance pathogen dispersal took place without there currently being any samples / genomes from (other) locations that could break up the very long branches in Fig. 3A, B. We consulted the World Animal Health Information System (WAHIS)[30] and the Scientific Committee on Antarctic Research (SCAR) databases[31] to look for unsampled outbreaks that would have been reported in the sub-antarctic area or in South Africa during the unsampled period – but none had been reported. These two long-term dispersal events could hence be due to gaps in current sampling efforts in the sub-Antarctic and Antarctic regions or to genomic data not being yet available at the time of our data analysis (but see the Discussion).

Finally, Fig. 3A shows that in the two clusters of sequences from the Southern Indian Ocean islands - one from the Crozet and one from the Kerguelen islands - mixtures of species can be observed, with southern elephant seals in each, but also king penguins, a brown skua, a kelp gull and a Cape petrel. This is in contrast with the other parts of the phylogeny, where each cluster is estimated to be host-specific, based on the currently available data. However, it is important to note that those host-specific clusters might result from a sampling bias rather than from host-specific transmission. In the South Georgia islands, differences between personal protective equipment (PPE) and staff training requirements for seal and bird carcass sampling led to much fewer seal carcasses being sampled than bird carcasses[22]. We attempted to avoid this sampling bias on Crozet and Kerguelen islands by having a more extensive sampling strategy, covering all species that were found dead at each sampling occasion, and with several individuals per species sampled at each site, when possible. This enabled us to detect multi-species clusters in our phylogeny, which was not detected previously (Fig. 3A). Importantly, this points to very frequent cross-species transmission, specifically from seals to scavengers (Fig. 1).

The presence of multi-species clusters may also be attributed to southern elephant seals spending time in close vicinity with high aggregation of seabirds on mixed-species colonies (Fig. 1), even though they do not feed on seabirds or marine mammals. This may have been critical for the inferred chain of transmission events (Fig. 3). Further, southern elephant seals undertake very long trips at sea over their life cycle[32], although their movements are much slower than those of birds, which may limit their ability to vector the virus, depending on how long an individual may stay infected. In their annual migration, they can move from South Georgia, Crozet or Kerguelen islands to the Antarctica coastal waters, several thousands of kilometres away, in several weeks[32]. Many of the movements are made at a limited range of latitudes, with possible overlap at the sole of the roaming areas of widely-spread seabird populations such as those of South Georgia, Crozet and Kerguelen islands.

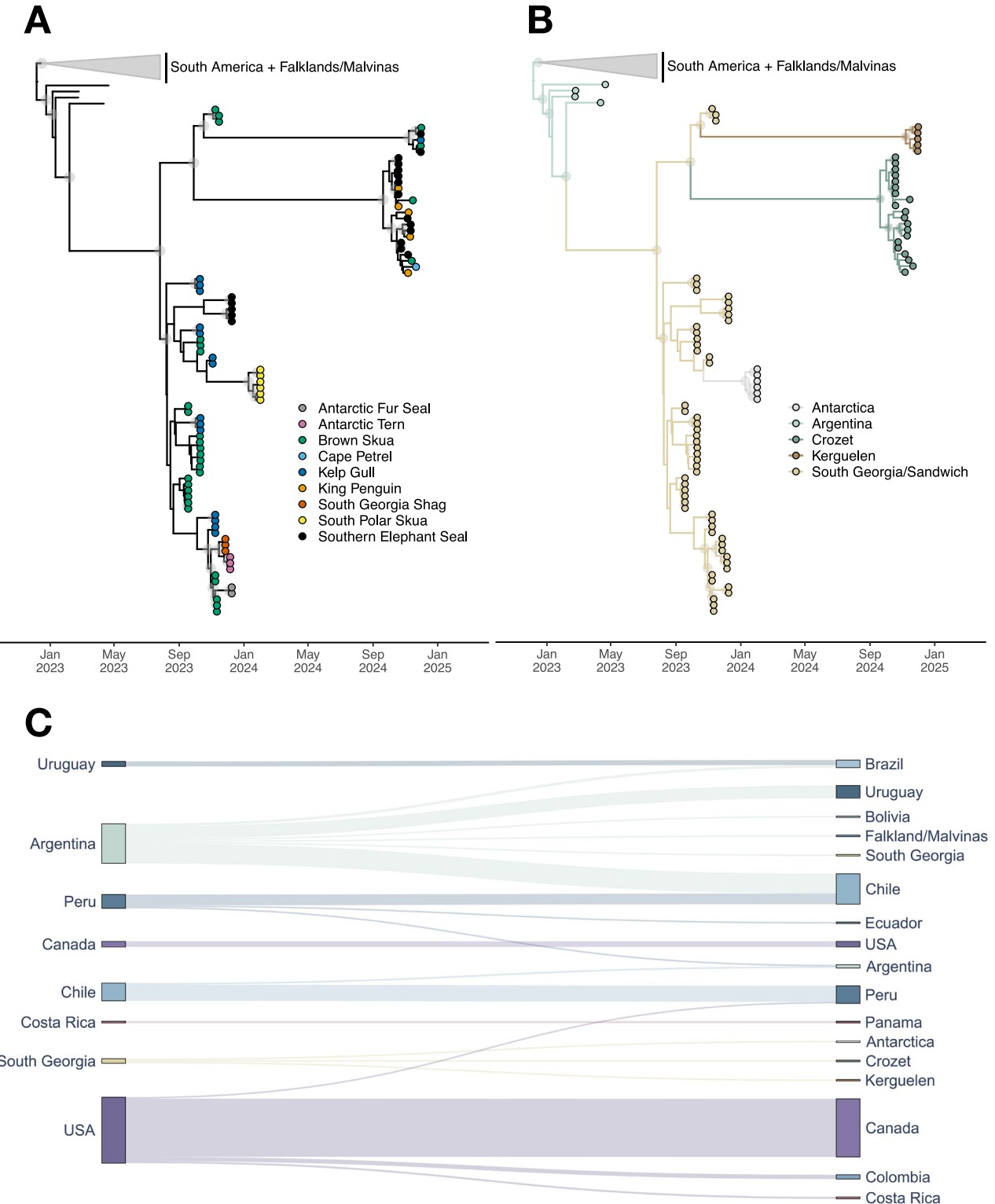

**Fig. 3 | Origin of the HPAI H5N1 2.3.4.4b infections on Crozet and Kerguelen islands.** All sequences from the Crozet and Kerguelen islands were obtained as part of this study. **A** Time-calibrated phylogeny based on near-complete genomes. Host species are annotated at the tips, and circles at the internal nodes show posterior support > 90%. **B** Location-annotated phylogeny based on whole concatenated genomes, showing all inferred ancestral locations. The inferred origin of the Crozet and Kerguelen island infections are the South Georgia islands. All ancestral locations inferred have 100% posterior support. **C** Sankey plot showing the number of transitions between locations through the estimated number of Markov jumps. The thickness of the lines are proportional to the number of Markov jumps from the location to the left into the location on the right, conditional on the corresponding Bayes factor being higher than 3. From an inferred single introduction into South Georgia Island from Argentina, we estimate HPAI H5N1 2.3.4.4b to have jumped once to Antarctica, Crozet and Kerguelen islands, based on currently available data. We refer to Extended Data Figs. 5 and 6 for additional visualisations. Color scheme according to ColorBrewer 2.0[74].

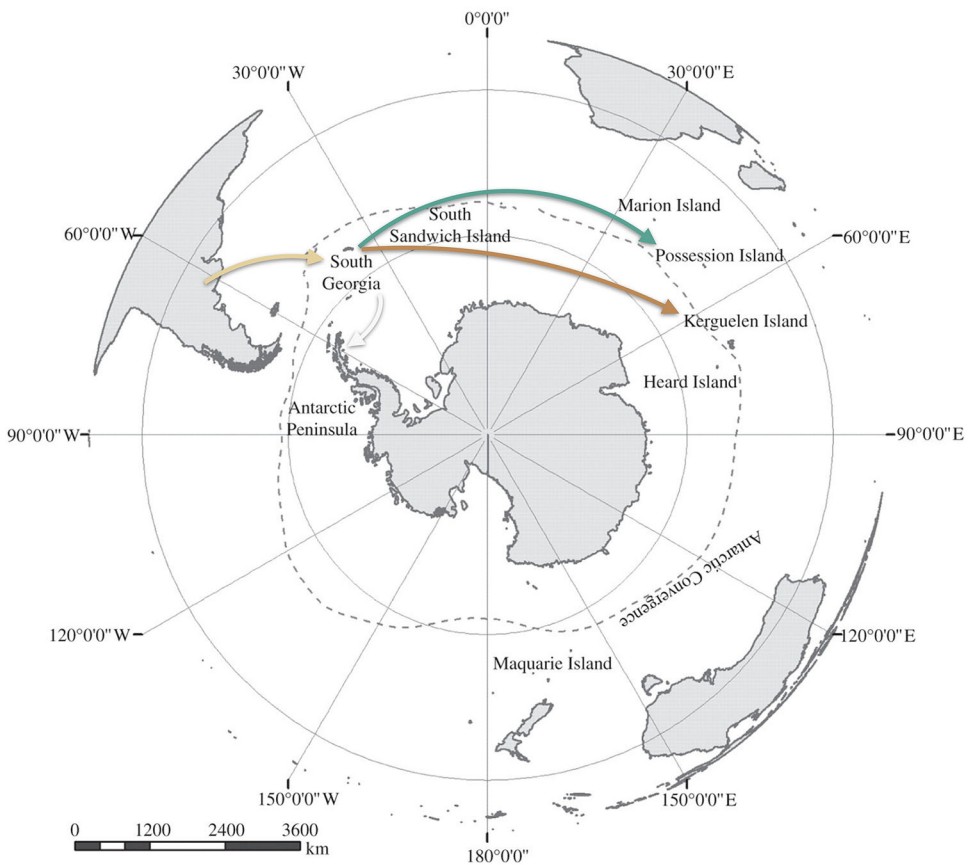

**Fig. 4 | Projection on a map of the phylogeographic analysis focused on our newly obtained sequences elucidates the spread of H5N1 around Antarctica, based on currently available data.** After having been imported into the South Georgia islands from Argentina, the latter served as the source for further dispersal of H5N1 across the sub-Antarctic region (Possession Island is one of the Crozet islands; all available sequences from South Georgia and the Sandwich Islands are actually from the South Georgia Islands). Note that the South Georgia islands are approximately 2100 kilometres east of Argentina, 5800 kilometres west from the Crozet islands and 6600 kilometres west from the Kerguelen islands. In turn, the Crozet and Kerguelen islands are respectively approximately 2500 and 3800 kilometres from South Africa, 2200 and 2000 kilometres from Antarctica and 5300 and 4000 kilometres from Australia. Map modified with permission from Mortimer et al.[75], with the same colour scheme as in Fig. 3B (arrows coloured according to destination; each arrow represents one introduction event).

## Molecular markers

The viral sequences obtained display certain molecular markers on the HA protein (S133A, S154N, T156A), which have been shown to increase binding to human-type receptors in vitro[33–36]. These markers have also been found in the majority of high pathogenicity H5 viruses since 2020. One of our sequences from a southern elephant seal analysed (A/southern_elephant_seal/Crozet/24P021415/2024) also features an E627K mutation in the PB2 protein, and another one from a king penguin (A/king_penguin/Crozet/24P021416/2024) features a mixed infection with E627K/E. The mutation E627K is considered to be one of the most important markers of adaptation to mammals, and has been identified in one other southern elephant seal sequence from the South Georgia islands[4]. However, our phylogenetic analysis (Fig. 3 and Extended Data Fig. 5, 6) does not show a monophyletic clustering of our (southern elephant seal) sequence from Crozet and Kerguelen islands with the southern elephant seal from the South Georgia islands, raising questions regarding the emergence and spread of this mutation. We will focus on unravelling the evolutionary relationship of this mutation in a future study, when more sequences from different sub-Antarctic islands have become available. We refer to Supplementary Data 1 for additional details on the other mutations that were investigated but not found in our sequences.

## Serological analyses of southern elephant seal samples

To explore whether the populations of southern elephant seals of Kerguelen had been exposed to H5 avian influenza viruses prior to the outbreak, we screened plasma samples of adult elephant seals collected in 2018, 2022 and 2023 using ELISA assays. We found no evidence for the presence of antibodies against protein H5 in the series of samples prior to 2024 (Fig. 5A, competitive ELISA; see Extended Data Fig. 9 for results with other ELISA assays). A few adults (n = 4) sampled in 2023 showed elevated H5-antibody levels, however, we cannot confidently interpret these values as H5-seropositive individuals. In 2024, we collected samples from 26 asymptomatic elephant seal pups after the outbreak had started, at a time when there were still many individuals dying. Among those sampled pups, Fig. 5B shows that two pups exhibited high levels of antibodies against the H5 protein, which suggests that they had been exposed and mounted a humoral immune response. The use of an indirect anti-H5 ELISA (Extended Data Fig. 9) and of a recently available multi-species competitive anti-H5 ELISA specifically validated for mammals confirmed that the two samples were positive. The sampling was cross-sectional, and it was not possible to determine whether those pups survived.

## Discussion

In addition to the pressing threats it poses to the local biodiversity, our detection of HPAI H5N1 clade 2.3.4.4.b on the isolated Crozet and Kerguelen archipelagos is of global concern as it shows the virus' ability to spread over exceptionally long distances and across isolated oceanic basins. Our results suggest that South Georgia is the origin of the introductions into both Crozet and Kerguelen islands. While all node supports within the tree topology are over 90%, the long branches

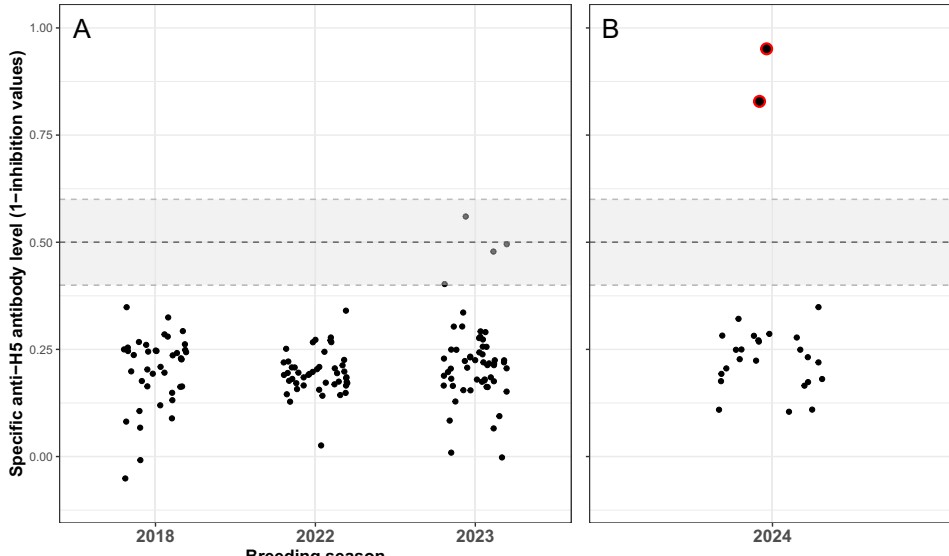

**Fig. 5 | Quantification of antibodies against the AIV-H5 protein.** Scatterplots showing anti-H5 antibody levels using a competitive ELISA. **A** from adult southern elephant seals sampled in Kerguelen in 2018, 2022 and 2023; (**B**) and from pups sampled in 2024 during the outbreak. The grey area displays the range for inconclusive H5-seropositivity results, according to the kit manufacturer. The positive samples are highlighted with a red circle. Two pups are H5-seropositive among the 26 that were sampled during the outbreak. The results of complementary ELISA assays are reported in Extended Data Fig. 9. The source data are available in Supplement Data 2.

depict a year-long unsampled history, and there is hence uncertainty about the existence of unsampled outbreaks that could have happened during that time. It is thus uncertain whether the virus spread multiple times from South Georgia Island to one or several unsampled locations, and later from these unsampled locations to Crozet and Kerguelen islands, or if the virus spread directly from South Georgia Island to Crozet and Kerguelen islands. In both cases, given how sparse the lands are in the Southern Ocean and how sparse the seabird and marine mammal breeding sites are on the Antarctic continent (except on the Antarctic Peninsula), this implies that the hosts can remain infectious for sufficiently long periods to cover thousands of kilometres. This is especially relevant for the sub-Antarctic region, where several archipelagos are to date thought to be free of HPAI but potentially at risk of exposure. Further dispersal across the Southern Ocean should hence be considered at high risk and would threaten many other wildlife populations, most notably in the Australia and New Zealand sub-Antarctic islands. Several species are known to migrate between Southern Indian Ocean islands and coastal areas of mainland New Zealand and Australia, including the brown skuas[37]. Hence, the risk of introduction of the virus to Australia and New Zealand should not be considered negligible.

The Indian Ocean sub-Antarctic islands are remote, hence, the chances of between-distant-island transmission events could be thought of as being low, but our results show that this is not the case, likely because of significant connections via the movements of the widely roaming vertebrates they are hosting. In the meantime, the virus is also suspected to have arrived on Marion Island, of the Prince Edwards islands (South Africa)[38], awaiting testing in February 2025[39], but it is not known whether it has been responsible for die-offs on other significant regional biodiversity hotspots, such as Est and Cochon Island (Extended Data Fig. 2), in the Crozet archipelago, the more remote parts of Kerguelen islands, or Heard and McDonnald islands (Australia). Several sub-Antarctic islands are classified as UNESCO World Heritage Sites[24] because of their environmental significance (Extended Data Fig. 2), unique pristineness and abundant but sensitive fauna, and it is especially worrying that HPAI virus H5N1 has reached those areas already at least twice independently from the South Atlantic subantarctic.

Evidence of between-mammal transmission has already been provided in North America[2,40] and between southern elephant seals and other pinnipeds in South America[18,40,41]. Given the similarity of mortality patterns with what happened in the Valdes Peninsula in South America, with mostly southern elephant seals dying, we think that mammal-to-mammal transmission between the southern elephant seals is very likely both in Crozet and Kerguelen islands. Vertical transmission has recently been shown as a mode of transmission of HPAI between pinnipeds[42]. We recurrently observed terrestrial scavengers like giant petrels and brown skuas feeding on carcasses (Fig. 1 and Extended Data Fig. 1), and they may hence be part of the transmission chain between individuals, groups, colonies and across broader spatial scales such as island and between islands. Scavenging by species on floating carcasses as we observed (Extended Data Fig. 7) may also be important. Differentiating exposure from susceptibility will be important to further understand the eco-epizoological dynamics, which should be possible via serology analyses coupled with demographic studies. Our serology results, although small-scale and stemming from the beginning of the outbreak, provide hope that a proportion of exposed southern elephant seals become immune and might survive the infection. However, given the limitations aforementioned, we invite to take those results as a first insight and as a call to conduct further investigation into the immunisation dynamics.

Southern elephant seals are known to occasionally travel over thousands of kilometres[40], but this takes several weeks. Further, the clade that was dominant in South America among marine mammals was not the one spreading to South Georgia Island, nor to the Indian Ocean, despite the fact that – as in South America – southern elephant seals were strongly affected on South Georgia, the Crozet and the Kerguelen islands. This may suggest that the seals are unlikely long-distance vectors of the virus. Conversely, circumpolar movements of *Procellariiformes* are well-documented[43] (Extended Data Fig. 8), notably eastward, following wind direction, and species like giant petrels can cover thousands of kilometres in a week[44]. Giant petrels, as scavengers and widely roaming species at high southern latitudes, are an obvious candidate species to have played a role in the spread of the virus between the South Georgia islands area and the Southern Indian

Ocean islands. These species have already been identified as a potential spreader of avian influenza virus[45]. Brown skuas and kelp gulls, as other well-known scavengers on marine mammals and seabird colonies, could also have been involved, but perhaps less likely for the large-scale eastward spread due to their more restricted movements and their winter migration that involves more north-south movements than eastward or westward movements at high latitude[46]. Migratory movements are the obvious type of movement that may have been involved in the spread of HPAI H5N1, but large-scale foraging movements of breeding scavengers and predators may also have played a role, as well as less directed movements of individuals not currently involved in breeding, such as pre-breeders[47]. The long-distance carriage of the virus from the southern Atlantic to the southern Indian could thus have happened through scavenging seabirds and not seals. This would imply that bird-to-seal spillovers initiated massive mortality of southern elephant seals on both the Crozet and the Kerguelen islands, and that the introduction on the islands could have happened earlier through scavenger seabirds. Bird-to-seal transmission events have happened several times in South America but are uncommonly reported[4,16,17]. The timing of mortalities on Kerguelen, delayed more than two months after the return of the southern elephant seals at their colony, is supportive of that introduction scenario.

## Methods

### Ethical issues and permits

Permits to handle animals were obtained from the French Ministry of Research and the Animal Experimental Research Authority (APAFIS #31773-2019112519421390 v4). Sampling was conducted under permits from Préfecture des TAAF (A-2018-126, A-2023-82, A-2023-159) after evaluation by the Conseil de l'Environnement Polaire and Conseil National de la Protection de la Nature. After the suspicion of HPAI, permits were granted by the Préfecture des TAAF to pursue eco-epidemiological fieldwork (A-2024-92, A-2024-93, A-2024-95, A-2024-144).

### Study areas

The Crozet (46°24′ S, 51°45′ E) and Kerguelen (49°15′ S, 69°10′ E) archipelagos are part of the French Southern and Antarctic Lands (TAAF), and together with the Prince Edwards islands (Marion and Prince Edwards, South Africa) and Heard and McDonald islands (Australia), they constitute the main Southern Indian Ocean islands, where millions of seabirds and marine mammals aggregate each year to breed (Extended Data Fig. 2). Situated near the oceanic polar front, these islands are exposed to windy and cold oceanic climates. They have no permanently settled inhabitants nor domestic animals. Approximately 60 (in the winter) to 150 (in the summer) people are usually present on Crozet and Kerguelen islands at any time, mainly made up of military personnel, officials, scientific researchers and support staff. The islands are serviced four times a year by the research and supply ship *RV Marion Dufresne*, from Réunion Island.

### Sample collection

Breeding colonies of seabirds and marine mammals on Possession Island, Crozet archipelago, and Kerguelen islands are regularly monitored as part of long-term research projects supported by the French Polar Institute (IPEV) and management activities of the French Southern Territories National Nature Reserve. As part of disease ecology research and surveillance, field personnel working with wildlife had been trained to contribute to the detection and reporting of abnormal mortalities that could be due to infectious diseases. After the first abnormal mortalities were detected and led to suspicion of HPAI emergence on Possession Island, Crozet archipelago, specific sampling was organised at the site of the first detections as well at other locations with aggregations of seabirds and pinnipeds (Fig. 2). Special biosecurity measures were taken to prevent risks of contributing to the

spread of infectious agents and exposure of personnel, according to international guidelines adapted to specific field conditions[48]. On Crozet, we reached the sampling sites on foot. Sites close to refuges enabled a longer presence and, therefore, a more complete sampling effort while applying appropriate biosecurity measures. The Baie Américaine refuge was closed for sanitary reasons (as it was very close to an elephant seal colony), and this site was particularly difficult to access because of the distance to the station (8 km) and the biosecurity constraints that needed to be respected. On Kerguelen, we accessed our sampling site on the north-eastern side of the Courbet Peninsula by helicopter.

At each site, we focused sampling on the freshest accessible carcasses, with care to limit disturbance to the local wildlife. We focused our sampling efforts on collecting swab samples from the brain - using contendant tools and sterile swabs - given prior reports of a tropism of the HPAI virus for the central nervous system in marine mammals[17], and from the oral cavity for birds. Nasal and rectal samples are difficult to collect due to scavenging. We stored the first series of swab samples, collected between October 20th and 27th, in the field in dry sterile microcentrifuge tubes, before freezing and storing them at − 80 °C within 6 h. Samples were then brought back to Réunion Island and transported to the French National Reference Laboratory for avian influenza at ANSES (Ploufragan). We collected a second series of swabs over the following four weeks. These samples were stored and transported in the same way, except samples collected in the most remote part of the island, where dry swabs were maintained at ambient temperature (circa 6 °C) for several days before being frozen at − 80 °C. On Kerguelen, we carried out the sampling of carcasses on December 1st and 2nd, 2024. In addition, we had previously collected a series of blood samples from adult southern elephant seals in Kerguelen in 2018 ($n = 41$), 2022 ($n = 46$) and 2023 ($n = 57$). We spinned the blood samples and separated the plasma within 24 h after collection with heparinized syringes. In 2024, in parallel to having sampled carcasses, we sampled blood from 26 asymptomatic elephant seal pups. Due to the timing of collection of those samples, concomitant with the ongoing mortality, the seroprevalence could not be used to infer the proportion of pups that had been infected among those that eventually survived the event. After centrifugation in the field, we stored the plasma samples at − 20 °C until serology analyses.

### Avian influenza antigenic testing

We used qualitative rapid immunochromatographic strip tests (BIO K 106, Dpifit AIV Ag, BioX) in the field to detect the potential presence of avian influenza viruses in swab samples. The test detects the presence of the NP protein of avian influenza viruses. The detection level is expected to be 104.8 DIE50 / mL. Given the expected high viremic load in the brains of dead animals, we expected brain samples to be excellent in-field indicators of HPAI-induced mortality. We reported a test as positive when the control and T bands came out visible, but note that all samples were subject to further qRT-PCR testing, validation and confirmation at the French National Reference Laboratory for avian influenza.

### Detection and sequencing of HPAI H5N1 clade 2.3.4.4b

We eluted each dry swab in 1 mL of minimal essential medium (MEM). We extracted RNA from 100 μl of the elution medium with the addition of a mixture of RLT buffer and β-mercaptoethanol for the lysis step. The following steps were carried out in line with the NucleoMag Vet (Macherey) extraction procedures using the KingFisher automated system. We performed detection of avian influenza H5 HP clade 2.3.4.4b by qRT-PCR according to the protocol recommended by the European Union Reference Laboratory for avian influenza virus (adapted from Naguib et al.[49] and previously used in Briand et al.[50]). For full genome sequencing, we amplified the eight viral segments by RT-PCR[51], followed by library preparation and sequencing on a NextSeq2000 sequencer (Illumina).

## Phylogenetic analysis

On December 26th, 2024, we downloaded all available H5N1 HA sequences as well as full genomes from GenBank[52] and GISAID[53,54]. We first removed duplicate entries and complemented the resulting data set with our 25 newly generated H5N1 2.3.4.4b sequences from the Crozet and Kerguelen archipelagos (Extended Data Table 1). We aligned the resulting HA dataset using MAFFT v7.490[55], followed by manual / visual inspection. This yielded an initial data set containing 15072 sequences. We then performed maximum-likelihood (ML) phylogenetic inference using IQ-TREE v2.2.2.6[56] with automated model selection using ModelFinder[57], which selected the GTR + F + I + R6 model according to the Bayesian Information Criterion. We performed 1000 bootstrap replicates using the ultrafast bootstrap approximation approach (UFBoot)[58,59]. From this large ML phylogeny, we extracted a large, well-supported (bootstrap support 94%) cluster that contained our 25 novel HPAI H5N1 2.3.4.4b sequences from the Crozet and Kerguelen archipelagos, to be used in subsequent analyses. Within this cluster, our 25 sequences were included within an 83-sequences clade that had 100% bootstrap support and contained sequences from South Georgia and the South Sandwich islands, as well as Antarctica.

We proceeded to construct a full-genome data set for this cluster to improve phylogenetic resolution in a subsequent Bayesian phylogenetic analysis. We followed the approach in Uhart et al.[17] in that such a workflow limits the time for H5N1 to accrue mutations and diversify, thereby limiting genetic diversity, but did not restrict ourselves to only complete genomes in order to reduce both geographic and genetic biases[60]. We removed reassortant sequences while constructing ML trees to assess temporal signal in this complete genome data set and remove outliers using TempEst[61]. This resulted in a final data set consisting of 1221 genomic sequences, which exhibited a clear temporal signal.

We continued to perform a Bayesian time-calibrated phylogenetic analysis on this data set using Markov chain Monte Carlo as implemented in BEAST v1.10.5[62], employing the BEAGLE high-performance computational library for computational efficiency to run on a powerful graphics processing unit[63]. We employed a non-parametric coalescent model[64,65] as the tree prior, a general time-reversible substitution model[66] with a discretized gamma distribution to accommodate among-site rate heterogeneity[67], and an uncorrelated relaxed clock with an underlying lognormal distribution[68]. We used the default priors in BEAST v1.10.5[62], including a conditional reference prior on the mean evolutionary rate[69]. We ran this analysis until all relevant effective sample sizes reached at least 200, as assessed in Tracer v1.7.3[70], and checked if independent replicate analyses converged to the same posterior distribution. Based on the sampled posterior phylogenetic trees from this analysis, we performed a Bayesian discrete phylogeographic analysis using an asymmetric continuous-time Markov chain model with Bayesian stochastic search variable selection[71]. The following locations were present in our 1221-taxa data set and used as discrete states: Antarctica, Argentina, Bolivia, Brazil, Canada, Chile, Colombia, Costa Rica, Crozet islands, Ecuador, Falkland islands, Guatemala, Honduras, Kerguelen islands, Panama, Peru, South Georgia islands, United States, and Uruguay. We used the highest independent posterior subtree reconstruction (HIPSTR) approach in TreeAnnotator X[29] to construct a time-calibrated consensus phylogeny, after removing 10% of the chain as burn-in.

## Identification of molecular markers with zoonotic potential or mammal adaptation

The identification of the main molecular markers already identified in the literature was integrated just after the workflow for determining the consensus sequences and automatically applied to each sample using the FluMut application[72]. The results are presented following the mature H5 nomenclature.

## Serological analysis

We collected blood samples from southern elephant seals using heparinized syringes and separated the plasma in the field by centrifugation, which we then stored at − 20 °C until further analysis. We placed plasma samples at 56 °C for 30 min before performing analyses in order to ensure inactivation of the H5 HP viruses contained in plasma samples and decomplementation of the plasma. We used commercially-available enzyme-linked immunosorbent assay (ELISA) kits to detect antibodies against avian influenza viruses in plasma samples. As the main assay to detect and quantify specific antibodies against H5-AIV protein in plasma samples, we used a competitive ELISA (ID Screen® Influenza H5 Antibody Competition, reference no. FLUACH5-5P, Innovative Diagnostics SARL, Grabels, France). We conducted the assay according to the kit instructions. We obtained plasma antibody titre by reading absorbance at 450 nm on a microplate reader (Tecan Infinite® 200 Pro; Tecan Group Ltd., Mannendorf, Switzerland). Anti-H5 AIV antibody levels in plasma samples are expressed through an inhibition value calculated as the ratio of optical densities (OD) of the samples and of the negative control (NC): $Inhibition_{sample} = OD_{sample}/OD_{NC}$. We measured the between and within-assay repeatability using inhibition values calculated from repeated sub-samples run on each plate-run (between-assay coefficient of variation: $4.4 \pm 3.7\%$, $n = 8$ samples; within-assay coefficient of variation: $1.7 \pm 1.1\%$, $n = 6$ samples). As confirmatory serological assays, we ran an indirect commercial ELISA kit made for poultry (ID Screen® Influenza H5 Antibody Indirect, reference no. FLUH5S-5P, Innovative Diagnostics SARL, Grabels, France), and another multispecies competitive ELISA kit that is validated by the manufacturer for mammals (ID Screen® Influenza H5 Antibody Competition 3.0 Multi-species, reference no. FLUACH5V3-5P, Innovative Diagnostics SARL, Grabels, France). We adapted the indirect ELISA by replacing the anti-bird secondary antibody with an anti-canine secondary antibody (Goat anti-Canine IgG (H + L) Secondary antibody HRP, ThermoFischer Scientific catalogue number A18763). We used a 1:2000 dilution. Anti-Canine IgG secondary antibody can be used to detect IgG in elephant seals[73]. The confirmatory competitive ELISA was made following the manufacturer instructions.

## Reporting summary

Further information on research design is available in the Nature Portfolio Reporting Summary linked to this article.

## Data availability

We have deposited the 25 sequences generated in this study into the GISAID[53,54] database with accession numbers EPI_ISL_19747197 to EPI_ISL_19747221, and into the publicly accessible Genbank database with accession numbers PV294985 to PV295184. We provide in Supplementary Data 1 detail of the mutation analyses, Supplementary Data 2 the serology analyses results, in Supplementary Data 3 the file to reproduce the BEAST analysis (with the alignment removed to comply with GISAID terms of use), in Supplementary Data 4 the complete phylogenetic tree of Fig. 3 in Nexus format, in Supplementary Data 5 the GISAID acknowledgement link and the DOI associated with our dataset, in Supplementary Data 6 the FASTA file with our newly obtained sequences, in Supplementary Data 7 the metadata associated with those sequences, in Supplementary Data 8 a table that maps the GISAID and Genbank accession numbers to the individual gene sequences, and in Supplementary Data 9 a table with the strain names, GISAID accession numbers and Genbank accession numbers.

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

## Acknowledgements

We acknowledge critical support from the French Polar Institute (IPEV ECOPATH-1151, CYCLELEPH-1201), ANR AAPG ECOPATHS project (ANR-21-CE35-0016), REMOVE_DISEASE project (ANR-21-BIRE-0006; Biodiversa + and Water JPI joint call for projects under the BiodivRestore ERA-NET Cofund GA N°101003777) and CNRS Ecologie & Environnement SEE-Life ECOPATH. We also acknowledge specific support from the Ceva Wildlife Research Fund since the beginning of the recent HPAI panzootic affecting wild species. We thank Martha I. Nelson for helpful discussions regarding the detection of reassortant sequences. G.B. acknowledges support from the Research Foundation - Flanders ("Fonds voor Wetenschappelijk Onderzoek - Vlaanderen," G0E1420N, G098321N), from the European Union Horizon 2023 RIA project LEAPS (grant agreement no. 101094685), and from the DURABLE EU4Health project 02/2023-01/2027, which is co-funded by the European Union (call EU4H-2021-PJ4) under Grant Agreement No. 101102733. We also acknowledge support from Zone Atelier Terres Australes et Antarctiques (ZATA) and Direction de l'Environnement of TAAF. We conducted serological analyses at the GEMEX (CEFE) platform. We thank the French Polar Institute (IPEV) and TAAF for long-term and reactive support. We thank Amandine Gamble, Nicolas Keck, Gregory Jouvion, Rozenn Le Net, Karin Lemberger, Célia Lesage, Maxime Amy and SAGIR/Office Français de la Biodiversité (OFB) network for having contributed to organise the training of field personnel for wildlife disease surveillance, which helped responding to HPAI emergence on Crozet and Kerguelen islands. We also thank Antoine Stier (IPEV 119), Lucia Llorente, Nathan Thenon, Blaise Raymond, Marina Oger, Sara Boucheron, Loïs Angelvin, Francesco Bonadonna (IPEV 354), Céline Lebohec (IPEV 137), Christophe Barbraud (IPEV 109), Karine Delord, Aude Noiret, Bastien Bauger, Maxime Amy, Pierre-Marie Borne and Alice Mistou for their help at various stages of the study. The "NextSeq™ 2000" sequencing system and the "ThinkSystem SR650 V3 servers", used to obtain and analyse the data, were funded by a European Union grant.

## Author contributions

Author contributions: T.Bo., A.C., G.B. and B.G. conceived the study; T.Bo., J.T. and M.L. led the field work; C.D.P., R.F., T.Br., C.G. and C.R.M.

contributed to field sampling; F.-X.B. and F.S. performed the molecular analyses; E.H. performed the sequencing data curation; G.B., F.-X.B., A.C., S.L.H. and B.G. performed the sequencing analyses; M.L. performed the serological analyses; A.C., T.Bo. and G.B. wrote the first version of the manuscript and all authors provided inputs.

## Competing interests

The authors declare no competing interests.
