## [Peer Review file · Nature Communications]

Circumpolar Spread of Avian Influenza H5N1 to Southern Indian Ocean islands

Corresponding Author: Dr Thierry Boulinier

Version 0:

Reviewer comments:

Reviewer #1

(Remarks to the Author)

In this study, 25 genomic sequences were obtained from H5N1 avian influenza viruses collected in two islands in the Indian Ocean. Surprisingly, the sequences are closely related to H5N1 viruses collected in Argentina and South Georgia, evidence of long-distance circumpolar spread in wild birds. The study is fairly straightforward and well written and presented, although there is a full year of missing data where we don't know where the virus was geographically, owing to a large sampling gap.

Major comments

1. Have you checked WAHIS or other sources to determine if there are known outbreaks or mass mortality events in the region (for example in South Africa or other islands) without sequence data? If so, could these be simulated in the tree (e.g., Worobey et al 2021).
2. It would be helpful to have a concise summary table in the main text Results with the % positive by species and island and date, etc.
3. Can you say more about the direction of spillover between seals, skuas, penguins, etc? You suggest that skuas are likely vectors for introduction onto the islands, but also appear to be recipients of virus from other species acquired during scavenging. Maybe a cartoon with hypotheses about the direction of transmission between these different species would clarify.

Minor comments

1. The tone is a little hyperbolic at times. For instance, could simplify the title to "Circumpolar spread of avian influenza H5N1 to southern Indian Ocean islands"
2. What time of year does the breeding season occur? January-Feb?
3. Most readers won't know what Procellariiformes means. Maybe put (petrels, shearwaters) in parentheses.
4. The GenBank accession numbers PV294985 - PV295184 should be included in the text and Data Availability sections.
5. "and other species in North America^{31,35}" this appears to be the wrong reference
6. Could you provide the complete tree in Figure 3, and XML, in a repository like GitHub?
7. Maybe use a combination of colors and shapes for the tips in Figure 3? It's hard to differentiate some of the colors.
8. GISAID is not considered a public database by Nature journals. Can you report GenBank accessions instead of GISAID accessions in Extended Data Table 1.

Reviewer #2

(Remarks to the Author)

Unprecedented and Devastating Circumpolar Spread of Avian Influenza H5N1

Clessin et al. provide a study documenting the detection of H5N1 HPAIV in the islands of Crozet and Kerguelen. This paper is of interest considering the importance of H5 HPAIV in these regions and potential for spread to other areas. Phylogenetic analysis shows the highest similarity to sequences from South Georgia, and the authors conclude that the virus has therefore come from South Georgia. However considering the long branch lengths which suggests unsampled ancestry, this is difficult to say with such certainty. The authors also show serological analysis of H5 antibodies in seal, finding two reactors in ELISA from 2 seals, however considering that only one ELISA test was used which is not validated on this species according to the manufacture it is difficult to know if these two reactors are genuine. Overall, the study was an interesting read and provides more important information, particularly sequencing information, building upon what exists in the Antarctic region to facilitate future analysis. This review also understands the limitations and difficulties which must come with sampling in these remote regions. However, there are a few points to address below.

1. The outbreak of... doesn't make grammatical sense.
2. Change highly pathogenic to high pathogenicity as per WOHAI recognised nomenclature.
3. Changes required to distinguish between HPAI (the disease) and HPAIV (the virus)
4. Language is often imprecise, partially when it relates to virology, and often requires more specific detail to clarify meaning. One example 'originating in Asia in 1996 it spread across the world in different waves.' It is not clear what 'it' is and 'what' originated in Asia in 1996? H5 avian influenza virus has been around much longer. I presume the authors mean the goose/Guangdong lineage H5 HPAIV first emerged in Asia in 1996 where it re-entered wild bird populations and spread across the world, diversity into distinct clades according to the HA.
5. There needs to be differentiation among the outbreaks described, within the goose/Guangdong lineage H5 HPAIV there is a high degree of diversity within the HA and even more in the internal genes segments, therefore this is not one virus. The authors describe outbreaks i.e. south Africa 2016-17, however this is not the same virus at the 2020-25 panzootic, despite having a common ancestor.
6. Some of the claims in the introduction are overstated compared to the limitations of the results. For example, in the intro it is stated that the article provides key insights into 'evidence of mammal-to-mammal transmission'. I see no evidence for this in the article.
7. I don't understand the first sentence in 'descriptive epidemiology'
8. More detail on the antigenic test was used saying it targets NP from influenza viruses.
9. Dates of sampling would be useful to be incorporated into Fig 2.
10. B3.2 in all samples, please current in the text for precision? I think the authors need more basic description of the virus from sequencing, confirming that it is H5N1 clade 2.3.4.4b genotype B3.2.
11. It is not clear where sequences from this paper sit in Fig3A
12. It is not clear if figure 3 represents HA analysis only, where phylogenetic trees constructed for all other gene segments, these don't appear to be displayed? Where phylogenetic trees constructed for all gene segments, and not concatenated.
13. Sequences from the Falkland islands are not included in Figure 3
14. What was the genome coverage, where full genomes produced for all gene segments for all sequences, this detail should be included in supplementary table.
15. Was analysis performed looking at within islands spread, there is reference to evidence of mammal-to-mammal transmission in the introduction, but this does not appear to have been presented.
16. There are some very large branch lengths from South Georgia, in addition there is a lack of recent south American sequences requested. Therefore, it reduces the likely introduction routes into Crozet and Kerguelen. I'm not convinced that from the data provided that it can be stated in the state that 'we show that... introductions of the virus... from south Georgia islands. The authors need to rephrase these claims with caveats above.
17. Sequence names should be included as well as accession numbers in extended data table 1
18. Which sequences contained the molecular markers
19. Which nomenclature is being used for HA mutations, complete H5, mature H5 or mature H3?
20. References for receptor binding mutations increasing receptor binding.
21. PB2 E627K can emerge after single infection events in single mammals, therefore it is unsurprising that it does not show a monophyletic relationship with other viruses containing PB2 E627K.
22. Figure number for FluMut analysis.
23. References are needed for the qRT-PCR assay or assays used in detection, was a single assay run which detections with diagnostic thresholds and Ct values for each assay which should be added to Extended Data table 1.
24. The ELISA used is only validated on domestic & wild avian species according to the manufacturer's website, was any validation run on seal sera, if not then the results should be confirmed by another serological tool, such as microneutralization or Hemagglutination inhibition assay.
25. Is Figure 4 based on BEAST analysis, does it have any statistical weighting on incursion routes?
26. The discussion needs to address some of the uncertainties in the result more directly, such as unsampled ancestry and likely introduction routes, serological findings etc.

Reviewer #3

(Remarks to the Author)

In this manuscript Clessen et al perform an in depth outbreak investigation on an H5N1 outbreak in avian and mammalian species in the subantarctic/southern Indian ocean islands of Crozet and Kerguelen. This is a fascinating and extremely important study, performing an investigation in an area of the world that is currently being impacted by these viruses, but has had very little analysis performed in it so far. This work begins to fill in a huge gap in the sequencing data in this remote, but incredibly biodiverse region, and gives many insights, not just into the disease outbreak, but more generally the ecology of the animals that live in this region of the world.

I believe this work is highly important and should be published – I have a few mostly textual suggestions to help improve its readability to, what I believe would be the diverse audience who would find it interesting. My main critique (although I understand why this is the case) is that at present some highly relevant mainland Antarctic sequences are not included in this analysis, which would alter some of the interpretation of the authors own phylogenetic analysis. I don't think its necessary to repeat all the analyses (as the authors say, these sequences are slowly dripping in at the moment so any redoing of analysis is likely to have to be done repeatedly as more sequences come in), however acknowledging their presence and making some comments in the discussion would satisfy me that these sequences, and the impact they will have on future analysis have been analysed and considered .

Major points

- One small issue I have with how the paper is written is the repeated suggestions that the virus got into Crozet and Kerguelen from South Georgia, although this is certainly possible, due to the very patchy surveillance in the subantarctic/southern Atlantic/southern Indian oceans and Antarctic I believe there are other possibilities, and that south Georgie and Cro/Ker could have been incursions from a shared source, rather than there being s definite directionality. I think it would be good for the authors to acknowledge this uncertainty and how gaps in surveillance can alter the interpretation.

Minor points:

I'm happy to defer to the editor but probably the title could be a little bit more descriptive (e.g. summerise the findings and including the location ' e.g. subantarctic southern Indian Ocean' or something similar.

Introduction:

This paper only discusses possible spread to the outlying subantarctic islands of Australia and New Zealand, not to their mainlands. If the authors do think outbreaks on CRO/KER pose an increased risk of H5N1 incursion to mainland Australia/NZ, they might want to explore this in the article- are there any possible incursion routes that they can identify (either direct or indirect) based on know bird migration routes?

Reference 7 seems to be the wrong paper – possibly was meant to cite a different paper by the same author e.g.

<https://pubmed.ncbi.nlm.nih.gov/31833671/>

'Wandering albatross' is no longer the recognised name for that bird species and should instead by 'snowy albatross' "Originating in Asia in 1996 5,6," – H5N1 did not arise in Asia in 1996, I believe the authors specifically meant the high pathogenicity Goose/Guangdong H5Nx lineage was originally sampled in 1996 (in China).

"it reached South Africa 7". The authors are talking about H5N1 here, but it was an H5N8 (of the goose Guangdong HA lineage) that reached south Africa, Again the language in the introduction could use some cleaning up to specify this is related to the H5HA, not more specifically H5N1.

"devastating effect on seal" – I believe most early mortalities were sea lions rather than seals (elephant seals and fur seals were later in Argentina) – maybe worth using a collective term for these mammals such as pinnipeds or semi-aquatic mammals.

Although some details are given, It might be worth including some more details of the location of these French Austral lands and seas, relative to mainly antarctica, south Africa and south Georgia (distance and direction in km) in the introduction.

"pressure from global change" – possibly the authors mean climate change, or global warming here specifically?

Results:

Any data on sex ratios, if available, could be good to include. Previous studies have found that some species have infection rates that bias in favour of a particular sex: <https://doi.org/10.1080/00063657.2024.2396563>. For elephant seals in particular this would be of interest as there are significant sex-related differences in feeding strategies, metabolism etc

Figure 3 – the colours are a little hard to read on panel A and B – could something like shaped also be included to make the figure clearer to read?

'Molecular markers' – Looking at the sequences there is also a second sequence with mixed E627E/K

(24P021416_H5N1_D-24-09077_HP_B3.2_CRO_2024-10-20). My feeling is that mixed mammalian adaptations usually implies direct spillover from an avian source (rather than sustained mammal to mammal transmission) as these mutation is arising de novo – It might be interesting to mention this second sequence and maybe investigate the proportion of 627E vs K reads within in.

The serology aspect of this paper is generally good but a bit rudimentary- given the limitations of the environment I suppose this is to be expected. Possibly one or more of the following might help develop this a bit further:

-Testing of samples (particularly ELISA-positive or inconclusive samples) with an additional assay, e.g. haemagglutination inhibition or neutralisation assay

-To what extent (if any) do the authors think cross-reactivity with seal influenza viruses might impact results?

-Why is serology only performed for elephant seals and not avian species- was this a deliberate study choice or a constraint of sampling?

Discussion – something absent in the discussion is the absence of the sustained mammalian-adapted B3.2 lineage (the one with PB2 D701N/Q591K, described in detail in Uhart et al). The absence of this virus, despite its presence in the falklands islands, and also in the mainland Antarctic, might suggest elephant seals are less likely to be the vectors of this virus to the southern Indian/subantarctic islands.

Also – although I understand the phylogenetic analysis was done prior to the handful of mainland Antarctic H5 sequences

being shared publically, it would still be good to mention where they would fall in the phylogenetic trees. Including them in some phylogenetics on the authors new sequences shows a few of these sequences fall fairly close to the south Indian ocean sequences, at least for a few of the segments, particularly the sequences from Torgersen Island (A/Brown_Skua/Torgersen_Island/o81-b82/2024|EPI_ISL_19645365, A/Brown_skua/Torgersen_Island/o8182/2024|EPI_ISL_19745586). These sequences could majorly change the interpretation of the phylogenetics, suggesting spread to Ker/Cro from the mainland Antarctic, rather than by the subpolar route.

Version 1:

Reviewer comments:

Reviewer #1

(Remarks to the Author)

The revised manuscript suitably addresses the reviewer comments.

Reviewer #2

(Remarks to the Author)

The authors have fully addressed or appropriately justified all of my comments on the manuscript, and I believe these revisions have significantly improved the overall quality of the work.

Reviewer #3

(Remarks to the Author)

After re-reviewing the paper, we are satisfied that our concerns about the initial submission have been adequately addressed. In particular we commend the authors on the new serology data included in this current version of the paper which we believe greatly increases the strength of the conclusions.

Reviewer #4

(Remarks to the Author)

REVIEWER COMMENTS

Reviewer #1 (Remarks to the Author):

In this study, 25 genomic sequences were obtained from H5N1 avian influenza viruses collected in two islands in the Indian Ocean. Surprisingly, the sequences are closely related to H5N1 viruses collected in Argentina and South Georgia, evidence of long-distance circumpolar spread in wild birds. The study is fairly straightforward and well written and presented, although there is a full year of missing data where we don't know where the virus was geographically, owing to a large sampling gap.

Response: We thank the Reviewer for this positive assessment of our work. We agree that there is a large (temporal and spatial) sampling gap in H5N1 avian influenza virus sequences since 2023 in the Antarctic and sub-Antarctic areas, which stresses even more the importance of the results we present.

Major comments

1. Have you checked WAHIS or other sources to determine if there are known outbreaks or mass mortality events in the region (for example in South Africa or other islands) without sequence data? If so, could these be simulated in the tree (e.g., Worobey et al 2021).

Response: We thank the Reviewer for the suggestion. We had indeed checked the WAHIS and the SCAR databases. No outbreaks were reported in South Africa during that period, nor in other islands and mainland Antarctica that would be in the area in between Crozet / Kerguelen and South Georgia. We now explicitly mention this in the Results section, line 207-210, with the sentence “We consulted the World Animal Health Information System (WAHIS)³⁰ and the Scientific Committee on Antarctic Research (SCAR) databases³¹ to look for unsampled outbreaks that would have been reported in the subantarctic area or in South Africa during the unsampled period – but none had been reported.”

2. It would be helpful to have a concise summary table in the main text Results with the % positive by species and island and date, etc.

Response: We thank the Reviewer for this suggestion. However, these details are already available in Extended Table 1, and we also show in Figure 2 which samples were collected from where, how many tested positive and how many were successfully sequenced. As such, we preferred to summarize our sampling results on maps (i.e., Figure 2) rather than in a summary table of prevalence by island and by species, because showing mortality events at the different locations is highly relevant. Given that our samples were all obtained from carcasses, we do not think it is insightful to provide prevalences in a summary table, because we do not want readers to think that prevalence in carcasses provides an indication of the mortality rates among the different species, nor that it is interesting to compare HPAI prevalence in carcasses between species and locations (e.g., Crozet / Kerguelen). Those prevalences actually do not really have biological significance and we do not want those to be misinterpreted. However, we acknowledge that it can be useful to provide them as a methodological reference, to illustrate that almost all samples coming from an influenza mass mortality event are expected to be HPAI-positive. We have hence inserted a sentence in which

the proportions of carcasses found positive are reported, highlighting that most carcasses from this mass mortality event share the same cause of death (line 173-178: “A very high proportion of the carcasses tested came out positive for H5N1 by qRT-PCR (overall: 91% [65/71]; 88% [32/36] and 100% [10/10] for the elephant seals on Crozet and Kerguelen islands, respectively, 92% (11/12) for the king penguins on Crozet islands, and 100% (7/7) for the brown skuas over both islands; we also found three kelp gulls and one Cape petrel to be qRT-PCR-positive for HPAI H5N1; see Extended Data Table 1 for details).”.

3. Can you say more about the direction of spillover between seals, skuas, penguins, etc? You suggest that skuas are likely vectors for introduction onto the islands, but also appear to be recipients of virus from other species acquired during scavenging. Maybe a cartoon with hypotheses about the direction of transmission between these different species would clarify.

Response: We agree with the Reviewer that those are critically important questions, and that these will be essential to explore in follow-up analyses that we are planning. However, with our current 25 sequences (from Crozet / Kerguelen), we cannot provide a more detailed insight into the between-species transmission dynamics as this would require more sequences from all the species affected, including from those that eventually did not die from the infection. We don't suggest, and actually don't think that skuas are a likely vector for the introductions onto the islands. Skuas are expected to do rather North-South migrations, while giant petrels are expected to do circumpolar movements (but see below our response to reviewer #3: implications of skuas for further spread within the Indian ocean is a real concern in our opinion). However, at this stage we prefer to not go into too hypothetical scenarios about the transmission dynamics between the different species, as our data would not provide strong support to this end.

Minor comments

1. The tone is a little hyperbolic at times. For instance, could simplify the title to “Circumpolar spread of avian influenza H5N1 to southern Indian Ocean islands”

Response: We agree with the Reviewer and are grateful for the title suggestion which we now have adopted as the title of our revised manuscript. We also replaced the sentence in the introduction mentioning evidence of mammal-to-mammal transmission by a sentence in the discussion to say that mammal-to-mammal transmission was evidenced in South America, and that in Crozet and Kerguelen, the mortality pattern with mostly mammals affected suggests that it is not different from what was reported in South America. Line 352-357 now reads: “Evidence of between-mammal transmission has already been provided in North America^{2,40} and between southern elephant seals and other pinnipeds in South America^{18,40,41}. Given the similarity of mortality patterns with what happened in the Valdes Peninsula in South America, with mostly southern elephant seals dying, we think that mammal-to-mammal transmission between the southern elephant seals is very likely both in Crozet and Kerguelen.”

2. What time of year does the breeding season occur? January-Feb?

Response: The breeding season for most species is between October and March. We have added this to the introduction, line 76-78: “They aggregate in large numbers for breeding on

Crozet and Kerguelen islands each year between October and March, along with hundreds of thousands of marine mammals ²⁴.”

3. Most readers won't know what Procellariiformes means. Maybe put (petrels, shearwaters) in parentheses.

Response: We agree with the Reviewer and have made the suggested change (line 164). Our updated sentence now reads: “We also observed scavenging at sea on floating carcasses by Procellariiformes (petrels, shearwaters, albatrosses) and brown skuas.”

4. The GenBank accession numbers PV294985 - PV295184 should be included in the text and Data Availability sections.

Response: We agree with the Reviewer and have made the suggested changes, the first sentence of the Data Availability section (line 399-402) now reads: “We have deposited the 25 sequences generated in this study into the GISAID ^{50,51} database with accession numbers EPI_ISL_19747197 to EPI_ISL_19747221, and into the publicly accessible Genbank database with accession numbers PV294985 to PV295184.”.

5. “and other species in North America^{31,35} this appears to be the wrong reference

Response: We thank the Reviewer for noticing that; the correct reference is <https://www.nature.com/articles/s41586-024-08054-z>, which we have corrected (line 353).

6. Could you provide the complete tree in Figure 3, and XML, in a repository like GitHub?

Response: We agree with the Reviewer that this will improve the repeatability of our results, and now provide the complete phylogenetic tree and the XML files as supplementary files to our manuscript. Note that we removed the actual sequence data from the XML file, to comply with GISAID's data usage policy.

7. Maybe use a combination of colors and shapes for the tips in Figure 3? It's hard to differentiate some of the colors.

Response: We agree with the Reviewer that some of the colors in Figure 3A were difficult to distinguish. We have changed the color panel of Figure 3A (Figure 3B looks fine to us, but we have increased the figure quality) to improve the readability of the figure, and have made the corresponding modifications to other figures as well to keep the consistency of the color scheme used throughout our manuscript.

8. GISAID is not considered a public database by Nature journals. Can you report GenBank accessions instead of GISAID accessions in Extended Data Table 1.

Response: We have now added the Genbank accession numbers but opted to also keep the GISAID accession numbers in our table. Since the GISAID database contains more influenza sequences, has higher quality metadata and is more convenient to use than Genbank, we think it is important for the visibility and reuse of our novel sequences to provide GISAID accession numbers as well.

Reviewer #2 (Remarks to the Author):

Unprecedented and Devastating Circumpolar Spread of Avian Influenza H5N1

Clessin et al. provide a study documenting the detection of H5N1 HPAIV in the islands of Crozet and Kerguelen. This paper is of interest considering the importance of H5 HPAIV in these regions and potential for spread to other areas. Phylogenetic analysis shows the highest similarity to sequences from South Georgia, and the authors conclude that the virus has therefore come from South Georgia. However considering the long branch lengths which suggests unsampled ancestry, this is difficult to say with such certainty. The authors also show serological analysis of H5 antibodies in seal, finding two reactors in ELISA from 2 seals, however considering that only one ELISA test was used which is not validated on this species according to the manufacturer it is difficult to know if these two reactors are genuine. Overall, the study was an interesting read and provides more important information, particularly sequencing information, building upon what exists in the Antarctic region to facilitate future analysis. This review also understands the limitations and difficulties which must come with sampling in these remote regions. However, there are a few points to address below.

Response: We thank the Reviewer for this positive assessment of our work.

1. The outbreak of... doesn't make grammatical sense.

Response: We agree with the Reviewer and have corrected the sentence, which now reads (line 46-49): "The ongoing outbreak of clade 2.3.4.4b of high pathogenicity avian influenza (HPAI) H5N1 virus has been unique not only in the large number of host species affected, with a wide range of both birds and mammals reported dead ¹⁻³, but also in the scale of the panzootic, with all continents except Oceania affected ⁴."

2. Change highly pathogenic to high pathogenicity as per WOAHP recognised nomenclature.

Response: We have made the requested change.

3. Changes required to distinguish between HPAI (the disease) and HPAIV (the virus)

Response: We have made the suggested changes.

4. Language is often imprecise, partially when it relates to virology, and often requires more specific detail to clarify meaning. One example 'originating in Asia in 1996 it spread across the world in different waves.' It is not clear what 'it' is and 'what' originated in Asia in 1996? H5 avian influenza virus has been around much longer. I presume the authors mean the goose/Guangdong lineage H5 HPAIV first emerged in Asia in 1996 where it re-entered wild bird populations and spread across the world, diversifying into distinct clades according to the HA.

Response: We thank the Reviewer for pointing out these imprecisions and have made changes accordingly, the corresponding section of the introduction now reads (line 49-57): “Originating in Asia in 1996 ^{5,6}, the A/Goose/Guangdong/1/1996 HPAIV lineage spread across the world in different waves, recombined and diversified into different clades of various H5Nx viruses. The clade 2.3.4.4b started to become a world-wide preoccupation in 2014, and in the large outbreak of 2016-2017 HPAI H5N8 viruses of the clade 2.3.4.4b reached South Africa ^{7,8}. Since 2021, HPAI H5N1 viruses of the clade 2.3.4.4b severely impacted wild bird species and poultry across Europe ⁹⁻¹¹, southern Africa ¹²⁻¹⁴, North America ¹⁵, and South America where it resulted in die-offs of tens of thousands of seabirds ¹⁶ and marine mammals ^{17,18}.”

5. There needs to be differentiation among the outbreaks described, within the goose/Guangdong lineage H5 HPAIV there is a high degree of diversity within the HA and even more in the internal genes segments, therefore this is not one virus. The authors describe outbreaks, i.e. south Africa 2016-17, however this is not the same virus at the 2020-25 panzootic, despite having a common ancestor.

Response: We thank the Reviewer for pointing out those imprecisions and have made changes accordingly (but see our response above to your previous comment).

6. Some of the claims in the introduction are overstated compared to the limitations of the results. For example, in the intro it is stated that the article provides key insights into ‘evidence of mammal-to-mammal transmission’. I see no evidence for this in the article.

Response: We agree with the Reviewer and have removed the mention of evidence of mammal-to-mammal transmission as we did not conduct an ancestral host reconstruction analysis. We do not have sufficient sequences from Crozet and Kerguelen to perform such an analysis. Instead, we have added a sentence in the discussion to say that mammal-to-mammal transmission was evidenced in South America, and that in Crozet and Kerguelen the mortality pattern with mostly mammals affected suggests that the situation there is not different. The sentence (line 352-357) reads: “Evidence of between-mammal transmission has already been provided in North America ^{2,40} and between southern elephant seals and other pinnipeds in South America ^{18,40,41}. Given the similarity of mortality patterns with what happened in the Valdes Peninsula in South America, with mostly southern elephant seals dying, we think that mammal-to-mammal transmission between the southern elephant seals is very likely both in Crozet and Kerguelen.”

7. I don’t understand the first sentence in ‘descriptive epidemiology’

Response: We have changed the sentence to improve clarity, and it now reads: “The abnormal die-offs of southern elephant seal pups on Possession Island, Crozet, were detected by one of us (CDP) in the third week of October 2024.” (line 126-128)

8. More detail on the antigenic test was used saying it targets NP from influenza viruses.

Response: We now repeat in the Results section that the antigenic test used targeted influenza A NP protein when we report its use on snowy albatrosses (line 154) and on all species (line 179 and line 663-664 in the caption of Extended Data Table 1). More details are provided on the target in the ‘Avian influenza antigenic testing’ part of the Methods section for

this information (line 508-516), which now reads: “We used qualitative rapid immunochromatographic strip tests (BIO K 106, Dpifit AIV Ag, BioX) in the field to detect the potential presence of avian influenza virus in swab samples. The test detects the presence of the NP protein of avian influenza viruses. The detection level is expected to be 104.8 DIE50 / mL. Given the expected high viremic load in the brains of dead animals, we expected brain samples to be excellent in-field indicators of HPAI induced mortality. We reported a test as positive when the control and T bands came out visible, but note that all samples were subject to further qRT-PCR testing, validation and confirmation at the French National Reference Laboratory for avian influenza.”

9. Dates of sampling would be useful to be incorporated into Fig 2.

Response: We thank the Reviewer for this suggestion. As the sampling dates stem from a relatively short time frame and include repeated visits to some sites, it is not convenient to report the dates on the figure. However, we now report in the Figure 2 legend that sampling was conducted between October 20th and November 24th 2024 on Crozet, and on December 1st 2024 on Kerguelen, and also specify that details are presented in Extended Table 1 (line 121-123).

10. B3.2 in all samples, please current in the text for precision? I think the authors need a more basic description of the virus from sequencing, confirming that it is H5N1 clade 2.3.4.4b genotype B3.2.

Response: Indeed, B3.2 for all positive samples. We have changed the first sentence of the phylogenetic analyses section to makes this more explicit (line 186-189): “All the viruses detected in our samples are H5N1, clade 2.3.4.4b, genotype B3.2, similarly to what was mostly detected in South America and in the south polar areas¹⁴.”

11. It is not clear where sequences from this paper sit in Fig3A.

Response: We thank the Reviewer for the comment. All sequences from Crozet and Kerguelen stem from the work done for this manuscript, this is now explicitly mentioned in the legend of Figure 3 (line 242-243): “Fig. 3 | Origin of the HPAI H5N1 2.3.4.4b infections on Crozet and Kerguelen. All sequences from Crozet and Kerguelen are generated as part of this study.”. The annotated phylogenetic trees in Fig. 3A and 3B have the same topologies, with Fig. 3B showing which sequences are from Crozet and Kerguelen (and Fig. 3A having the same sequences at the same height).

12. It is not clear if figure 3 represents HA analysis only, were phylogenetic trees constructed for all other gene segments, these don't appear to be displayed? Were phylogenetic trees constructed for all gene segments, and not concatenated.

Response: As there is no recombination, and as indicated in the Methods section, we used full genomes for the phylogenetic analyses presented in Fig. 3. We have added this precision in the legend of Fig. 3, which now reads “A) Time-calibrated phylogeny based on near-complete genomes. Host species are annotated at the tips, and circles at the internal nodes show posterior support >90%; B) Location-annotated phylogeny based on whole concatenated genomes, showing all inferred ancestral locations. The inferred origin of the

Crozet and Kerguelen island infections are the South Georgia Islands. All ancestral locations inferred have 100% posterior support". (line 243-248)

13. Sequences from the Falkland islands are not included in Figure 3.

Response: Sequences from the Falkland Islands are actually included in the analyses presented in Fig. 3, but because they are distantly related to our sequences of interest, they are collapsed within the "South America" group of branches - which we have now renamed "South America + Falklands/Malvinas" to make this clear.

14. What was the genome coverage, were full genomes produced for all gene segments for all sequences, this detail should be included in a supplementary table.

Response: We thank the Reviewer for the suggestion and have added these details in a new table, i.e. Extended Data Table 2.

15. Was analysis performed looking at within islands spread, there is reference to evidence of mammal-to-mammal transmission in the introduction, but this does not appear to have been presented.

Response: The Reviewer is correct. We have now removed the mention of evidence of mammal-to-mammal transmission as we did not conduct an ancestral host reconstruction analysis, because we do not have sufficient sequences from Crozet and Kerguelen to do so. Instead, we have added a sentence in the discussion to say that mammal-to-mammal transmission was evidenced in South America, and that in Crozet and Kerguelen, the mortality pattern with mostly mammals affected suggests that it is not different. Line 352-357: "Evidence of between-mammal transmission has already been provided in North America ^{2,40} and between southern elephant seals and other pinnipeds in South America ^{18,40,41}. Given the similarity of mortality patterns with what happened in the Valdes Peninsula in South America, with mostly southern elephant seals dying, we think that mammal-to-mammal transmission between the southern elephant seals is very likely both in Crozet and Kerguelen."

16. There are some very large branch lengths from South Georgia, in addition there is a lack of recent South American sequences requested. Therefore, it reduces the likely introduction routes into Crozet and Kerguelen. I'm not convinced that from the data provided that it can be stated in the state that 'we show that... introductions of the virus... from south Georgia islands. The authors need to rephrase these claims with caveats above.

Response: We thank the Reviewer for the comment. As the Reviewer will have noticed, we have in fact been careful in our phrasing throughout the manuscript, and clearly state multiple times that our results are based on currently available data. Further, we would like to point out that the South Georgia part of the tree is densely sampled, that all node supports are strong (>90%), and that the ancestral locations are inferred with 100% posterior support. Hence, independent introductions to Crozet, Kerguelen and South Georgia from a common source is an unlikely scenario in our opinion, because it would imply that this densely sampled South Georgia branch results from two different introductions rather than one, which is unlikely given how dense this section of the tree is. This would also go against what has been published in

Banyard *et al.* “Detection and spread of high pathogenicity avian influenza virus H5N1 in the Antarctic Region” (*Nat. Commun.* 15, 7433; 2024).

In our opinion, the uncertainty is rather that, because the Crozet and Kerguelen branches are long, there could have been multiple dispersal events to one or several unsampled locations from South Georgia, followed by multiple dispersal events from these unsampled locations to Crozet and Kerguelen. In other words, we consider the tree topology to be stable and trustworthy and that the South Georgia branch is sampled densely enough to be confident about the fact that it results from a unique introduction. This does not exclude at all that other introductions to South Georgia might have happened, but then those other introductions would have resulted in other unsampled branches, and hence would not change the branching topology of the Crozet and Kerguelen sequences. So in our opinion, there is indeed not much doubt about the fact that the infections on Crozet and Kerguelen originate from South Georgia, but there is strong uncertainty about eventual unsampled locations in between. Again though, we emphasize that our manuscript contains repeated mentions of our inference results being based on currently available data.

Nevertheless, we have added the following sentences to the main text, in the discussion (line 321-331): “Our results suggest that South Georgia Island is the origin of the introductions into both Crozet and Kerguelen. While all node supports within the tree topology are over 90%, the long branches depict a year long unsampled history, and there is hence uncertainty about the existence of unsampled outbreaks that could have happened during that time. It is thus uncertain whether the virus spread multiple times from South Georgia Island to one or several unsampled locations, and later from these unsampled locations to the Crozet Islands and the Kerguelen Islands, or if the virus spread directly from South Georgia Island to the Crozet Islands and the Kerguelen Islands. In both cases, given how sparse the lands are in the Southern Ocean and how sparse the seabird and marine mammal breeding sites are on the Antarctic continent (except on the Antarctic peninsula), this implies that the hosts can remain infectious for sufficiently long periods to cover thousands of kilometers. “

17. Sequence names should be included as well as accession numbers in extended data table 1.

Response: We thank the Reviewer for the suggestion and have updated Extended Data Table 1 accordingly.

18. Which sequences contained the molecular markers

Response: We have added the sequence names in the main text (line 276-279): “One of our sequences (A/southern_elephant_seal/Crozet/24P021415/2024) from a southern elephant seal analysed also features an E627K mutation in the PB2 protein, and another one from a king penguin (A/king_penguin/Crozet/24P021416/2024) features a mixed infection with E627K/E.”

19. Which nomenclature is being used for HA mutations, complete H5, mature H5 or mature H3?

Response: The HA nomenclature was based on the mature H5. We have added this information to the Methods section, line 576-577: “The results are presented following the mature H5 nomenclature.”

20. References for receptor binding mutations increasing receptor binding.

Response: We thank the Reviewer for pointing out this omission and have added the following references for receptor-binding mutations increasing receptor binding (line 275):

- **Suttie A, Deng YM, Greenhill AR, Dussart P, Horwood PF, Karlsson EA.** Inventory of molecular markers affecting biological characteristics of avian influenza A viruses. *Virus Genes*. 2019 Dec;55(6):739-768. doi: 10.1007/s11262-019-01700-z. Epub 2019 Aug 19. PMID: 31428925; PMCID: PMC6831541.
- **Yang ZY, Wei CJ, Kong WP, Wu L, Xu L, Smith DF, Nabel GJ.** Immunization by avian H5 influenza hemagglutinin mutants with altered receptor binding specificity. *Science*. 2007 Aug 10;317(5839):825-8. doi: 10.1126/science.1135165. PMID: 17690300; PMCID: PMC2367145.
- **Wang W, Lu B, Zhou H, Suguitan AL Jr, Cheng X, Subbarao K, Kemble G, Jin H.** Glycosylation at 158N of the hemagglutinin protein and receptor binding specificity synergistically affect the antigenicity and immunogenicity of a live attenuated H5N1 A/Vietnam/1203/2004 vaccine virus in ferrets. *J Virol*. 2010 Jul;84(13):6570-7. doi: 10.1128/JVI.00221-10. Epub 2010 Apr 28. PMID: 20427525; PMCID: PMC2903256.
- **Gao Y, Zhang Y, Shinya K, Deng G, Jiang Y, Li Z, Guan Y, Tian G, Li Y, Shi J, Liu L, Zeng X, Bu Z, Xia X, Kawaoka Y, Chen H.** Identification of amino acids in HA and PB2 critical for the transmission of H5N1 avian influenza viruses in a mammalian host. *PLoS Pathog*. 2009 Dec;5(12):e1000709. doi: 10.1371/journal.ppat.1000709. Epub 2009 Dec 24. PMID: 20041223; PMCID: PMC2791199.

21. PB2 E627K can emerge after single infection events in single mammals, therefore it is unsurprising that it does not show a monophyletic relationship with other viruses containing PB2 E627K.

Response: We thank the Reviewer for this comment. Given that the vast majority of individuals dying in South Georgia and in Crozet / Kerguelen were mammals, we could have expected the mutation to be fixed already in South Georgia, or at least to reach a very high prevalence. It comes as a surprise that the mutation did not invade the population and did not even reach a high prevalence (although more sequences are required to ascertain that). And thus, the fact that this happened not only in South Georgia, but now again in Crozet / Kerguelen, is even more intriguing. This might be an indication that the virus is successively infecting birds and mammals, which could prevent the mutation from being strongly selected and induce some sort of balancing selection. We refrain from discussing this in depth in the main manuscript text though, as it is far from our results and requires a lot more investigation (which was the point we wanted to emphasize).

22. Figure number for FluMut analysis.

Response: Given the size of the table, we have opted for a supplementary file rather than a figure. Hence the results of the FluMut analysis are presented in Supplementary File 1. Reference to Supplementary File 1 is now explicit in the result section (line 287-289): “We

refer to the Supplementary File 1 for additional details on the other mutations that were investigated but not found in our sequences.”

23. References are needed for the qRT-PCR assay or assays used in detection; was a single assay run which detections with diagnostic thresholds and Ct values for each assay which should be added to Extended Data table 1.

Response: We thank the Reviewer for the comment and suggestions. We ran one single duplex assay with 2 targets on the H5 gene. We have now added the Ct values to Extended Data Table 1. We did not threshold the Ct values and considered a sample to be positive as soon as at least one of the 2 targets was positive. We consider that setting a threshold would constitute an arbitrary decision, and would essentially not be useful since having false positive results is just as bad as having false negative ones in this specific situation (because the individual diagnostic is of no interest and the presence of HPAI is certain).

24. The ELISA used is only validated on domestic & wild avian species according to the manufacturer's website, was any validation run on seal sera, if not then the results should be confirmed by another serological tool, such as microneutralization or Hemagglutination inhibition assay.

Response: The reported ELISA results in Figure 5 are indeed from a competitive anti-H5 ELISA validated for avian species, and they suggest that two pups were seropositive against AIV H5. As other serological tools, we have now run (i) an indirect commercially available ELISA for poultry that we modified by using an anti-canine secondary antibody, and (ii) a recently made available multispecies competitive anti-H5 ELISA validated for birds and mammals (ID Screen Influenza H5 Antibody Competition 3.0 Multi-species), which both confirmed the results. We provide those confirmatory results in Extended Data Figure 9 (see the figure below) as well as the associated methodological details in the Methods section (lines 596-606): “As confirmatory serological assays, we ran an indirect commercial ELISA kit made for poultry (ID Screen® Influenza H5 Antibody Indirect, reference no. FLUH5S-5P, Innovative Diagnostics SARL, Grabels, France), and another multispecies competitive ELISA kit that is validated by the manufacturer for mammals (ID Screen® Influenza H5 Antibody Competition 3.0 Multi-species, reference no. FLUACH5V3-5P, Innovative Diagnostics SARL, Grabels, France). We adapted the indirect ELISA by replacing the anti-bird secondary antibody with an anti-canine secondary antibody (Goat anti-Canine IgG (H+L) Secondary antibody HRP, ThermoFischer Scientific catalogue number A18763). We used 1:2000 dilution. Anti-Canine IgG secondary antibody can be used to detect IgG in elephant seals ⁷³. The confirmatory competitive ELISA was made following the manufacturer instructions.”

Extended Data Fig. 9 | Competitive and indirect ELISA for H5 proteins performed on southern elephant seal plasmas. Top left plot is a reproduction of Figure 5 for comparison purposes. The grey area displays the range for inconclusive H5-seropositivity results, according to the kit manufacturer. A jitter was introduced to displace the data points horizontally to improve their visibility. Below right plot shows the correlation between the v2 and v3 multispecies H5 ELISA. The results confirm that two of the pup samples were seropositive against H5 AIV protein in 2024, with no evidence of exposure to H5 AIV prior to that season.”

25. Is Figure 4 based on BEAST analysis, does it have any statistical weighting on incursion routes?

Response: Figure 4 is a projection on a map of the long-distance transmission events shown in Figure 3B. Hence, Figure 4 is based on the Bayesian phylogeographic analysis, and as indicated in the Figure 3 legend, the ancestral locations are inferred with 100% posterior support (but see also Extended Data Figure 6). As mentioned repeatedly throughout our manuscript, our inferences are based on currently available data and unsampled locations are necessarily a limitation. We changed the title of the figure to make it more explicit that this is a different representation of Figure 3B, but then on a geographic map. The figure title is now “Projection on a map of the phylogeographic analysis focused on our newly obtained sequences, that elucidates the spread of H5N1 around Antarctica, based on the currently available data.” (line 258-260)

26. The discussion needs to address some of the uncertainties in the result more directly, such as unsampled ancestry and likely introduction routes, serological findings etc.

Response: We thank the Reviewer for the suggestion and have modified the discussion to explain the uncertainties brought about by the unsampled ancestry. The sentences related to the phylogeny added to the discussion are (line 321-331): “Our results suggest that South Georgia Island is the origin of the introductions into both the Crozet Islands and the Kerguelen Islands. While all node supports within the tree topology are over 90%, the long branches depict a year long unsampled history, and there is hence uncertainty about the existence of unsampled outbreaks that could have happened during that time. It is thus uncertain whether the virus spread multiple times from South Georgia Island to one or several unsampled locations, and later from these unsampled locations to the Crozet Islands and the Kerguelen Islands, or if the virus spread directly from South Georgia Island to the Crozet Islands and the Kerguelen Islands. In both cases, given how sparse the lands are in the Southern Ocean and how sparse the seabird and marine mammal breeding sites are on the Antarctic continent (except on the Antarctic peninsula), this implies that the hosts can remain infectious for sufficiently long periods to cover thousands of kilometers.”

Regarding the serology results, we have added the following (line 365-369): “Our serology results, although small-scale and stemming from the beginning of the outbreak, provide hope that a proportion of exposed southern elephant seals become immune and might survive the infection. However, given the limitations aforementioned, we invite to take those results as a first insight and as a call to conduct further investigation into the immunization dynamics.”

Reviewer #3 (Remarks to the Author):

In this manuscript Clessin et al perform an in depth outbreak investigation on an H5N1 outbreak in avian and mammalian species in the subantarctic/southern Indian ocean islands of Crozet and Kerguelen. This is a fascinating and extremely important study, performing an investigation in an area of the world that is currently being impacted by these viruses, but has had very little analysis performed in it so far. This work begins to fill in a huge gap in the sequencing data in this remote, but incredibly biodiverse region, and gives many insights, not just into the disease outbreak, but more generally the ecology of the animals that live in this region of the world.

Response: We thank the Reviewer for these very nice words.

I believe this work is highly important and should be published – I have a few mostly textual suggestions to help improve its readability to, what I believe would be the diverse audience who would find it interesting. My main critique (although I understand why this is the case) is that at present some highly relevant mainland Antarctic sequences are not included in this analysis, which would alter some of the interpretation of the authors own phylogenetic analysis. I don't think it's necessary to repeat all the analyses (as the authors say, these sequences are slowly dripping in at the moment so any redoing of analysis is likely to have to be done repeatedly as more sequences come in), however acknowledging their presence and

making some comments in the discussion would satisfy me that these sequences, and the impact they will have on future analysis have been analysed and considered.

Response: We agree with the Reviewer that having these sequences - which were made available **after** we downloaded all available sequences at the start of our phylogenetic analyses (on December 26th, 2024) - in our workflow would be very interesting. We also thank the Reviewer for acknowledging that this would involve redoing our time-consuming phylogenetic and phylodynamic analysis, and that this may even have to be repeated while we are working on a revised version of our manuscript. To the best of our knowledge, the Reviewer is referring to the following samples:

- two sequences from Torgersen Island with collection dates on December 17th and an unknown day in December, 2024 and corresponding submission dates of January 6th, 2025, and of February 23rd, 2025; both were sampled from brown skuas
- four recently submitted sequences from Antarctica with collection dates on December 10th, 25th and 26th, 2024, all with submission date of April 24th, 2025, and all samples from brown skuas

The collection dates of these samples hence do **not** decrease the large sequencing gap that we show in our work, as they constitute more recent samples than the ones we collected from the Crozet and Kerguelen Islands. However, we obviously do acknowledge that the source of introductions of these samples would be of great interest. Further, and this poses a challenging ethical topic, we do not wish to claim (nor scoop) novel insights based on other researchers' samples and corresponding genomic sequences, which they may currently have submitted for publication as well.

Regardless, we have performed an updated phylogenetic (but not phylogeographic) analysis that includes these six additional sequences, and we disclose the results here for the Reviewer(s). The figure below (with the modified color scheme mentioned earlier) clearly shows that these six sequences (denoted 'Additional Sequences') do not have any impact on the findings reported in our manuscript as they also originated from South Georgia but through an independent introduction. These six sequences are hence not linked to the introductions into Crozet and Kerguelen (i.e. they did not source the introductions into Crozet and Kerguelen, nor do they originate from Crozet and Kerguelen). As such, and based on our previously mentioned ethical considerations, we opt not to include these sequences in our results.

Major points

- One small issue I have with how the paper is written is the repeated suggestions that the virus got into Crozet and Kerguelen from South Georgia, although this is certainly possible,

due to the very patchy surveillance in the subantarctic/southern Atlantic/southern Indian oceans and Antarctic I believe there are other possibilities, and that south Georgia and Cro/Ker could have been incursions from a shared source, rather than there being a definite directionality. I think it would be good for the authors to acknowledge this uncertainty and how gaps in surveillance can alter the interpretation.

Response: We thank the Reviewer for the comment, and have now added a sentence to emphasize the limitations induced by the unsampled ancestry. As the Reviewer will have noticed, we have in fact been careful in our phrasing throughout the manuscript, and clearly state multiple times that our results are based on currently available data. Further, we would like to point out that the South Georgia part of the tree is densely sampled, that all node supports are strong (>90%), and that the ancestral locations are inferred with 100% posterior support. Hence, independent introductions to Crozet, Kerguelen and South Georgia from a common source is an unlikely scenario in our opinion, because it would imply that this densely sampled South Georgia branch results from two different introductions rather than one, which is unlikely given how dense this section of the tree is. This would also go against what has been published in Banyard *et al.* "Detection and spread of high pathogenicity avian influenza virus H5N1 in the Antarctic Region" (*Nat. Commun.* 15, 7433; 2024).

In our opinion, the uncertainty is rather that because the Crozet and Kerguelen branches are long, there could have been multiple dispersal events to one or several unsampled locations from South Georgia, followed by multiple dispersal events from these unsampled locations to Crozet and Kerguelen. In other words, we consider the tree topology to be stable and trustworthy and that the South Georgia branch is sampled densely enough to be confident about the fact that it results from a unique introduction. This does not exclude at all that other introductions to South Georgia might have happened, but then those other introductions would have resulted in other unsampled branches, and hence would not change the branching topology of the Crozet and Kerguelen sequences. So in our opinion, there is indeed not much doubt about the fact that the infections on Crozet and Kerguelen originate from South Georgia, but there is strong uncertainty about eventual unsampled locations in between. Again though, we emphasize that our manuscript contains repeated mentions of our inference results being based on currently available data.

We added some sentences in the discussion to provide more details into those uncertainties (line 321-331): "Our results suggest that South Georgia Island is the origin of the introductions into both the Crozet Islands and the Kerguelen Islands. While all node supports within the tree topology are over 90%, the long branches depict a year long unsampled history, and there is hence uncertainty about the existence of unsampled outbreaks that could have happened during that time. It is thus uncertain whether the virus spread multiple times from South Georgia Island to one or several unsampled locations, and later from these unsampled locations to the Crozet Islands and the Kerguelen Islands, or if the virus spread directly from South Georgia Island to the Crozet Islands and the Kerguelen Islands. In both cases, given how sparse the lands are in the Southern Ocean and how sparse the seabird and marine mammal breeding sites are on the Antarctic continent (except on the Antarctic peninsula), this implies that the hosts can remain infectious for sufficiently long periods to cover thousands of kilometers."

Minor points:

I'm happy to defer to the editor but probably the title could be a little bit more descriptive (e.g. summarize the findings and include the location 'e.g. subantarctic southern Indian Ocean' or something similar.

Response: As per Reviewer #1's suggestion, we have changed our title to "Circumpolar spread of avian influenza H5N1 to southern Indian Ocean islands".

Introduction:

This paper only discusses possible spread to the outlying subantarctic islands of Australia and New Zealand, not to their mainlands. If the authors do think outbreaks on CRO/KER pose an increased risk of H5N1 incursion to mainland Australia/NZ, they might want to explore this in the article- are there any possible incursion routes that they can identify (either direct or indirect) based on know bird migration routes?

Response: We thank the Reviewer for the suggestion. We initially did not include this in the discussion as we think this is hypothetical, however we agree that given the high impact such an introduction would have, it is indeed relevant to mention the risk. We have hence added the following sentence to the discussion (line 335-338): "Several species are known to migrate between Southern Indian Ocean islands and coastal areas of mainland New Zealand and Australia, including the brown skuas ³⁹. Hence, the risk of introduction of the virus to Australia and New Zealand should not be considered negligible.", and cited this preprint: <https://www.researchsquare.com/article/rs-5874357/v1>. Even though it is a preprint, we think their Figure 2 (shown below for the Reviewer's convenience), depicting tracking data of brown skuas obtained with geolocators, is a really convincing illustration of the risk.

Reference 7 seems to be the wrong paper – possibly was meant to cite a different paper by the same author e.g. <https://pubmed.ncbi.nlm.nih.gov/31833671/>

Response: We thank the Reviewer for noticing this. We have changed the citation to: <https://bioone.org/journals/avian-diseases/volume-63/issue-sp1/11869-042518-Reg.1/The-Incursion-and-Spread-of-Highly-Pathogenic-Avian-Influenza-H5N8/10.1637/11869-042518-Reg.1.short>.

'Wandering albatross' is no longer the recognised name for that bird species and should instead be 'snowy albatross'

Response: We thank the Reviewer for noticing this, and we have made changes to our manuscript accordingly.

"Originating in Asia in 1996 5,6," – H5N1 did not arise in Asia in 1996, I believe the authors specifically meant the high pathogenicity Goose/Guangdong H5Nx lineage was originally sampled in 1996 (in China).

Response: We thank the Reviewer for pointing out those imprecisions and modified the first paragraph of the introduction accordingly, line 49-57: "Originating in Asia in 1996 ^{5,6}, the A/Goose/Guangdong/1/1996 HPAIV lineage spread across the world in different waves, recombined and diversified into different clades of various H5Nx viruses. The clade 2.3.4.4b started to become a world-wide preoccupation in 2014, and in the large outbreak of 2016-2017 HPAI H5N8 viruses of the clade 2.3.4.4b reached South Africa ^{7,8}. Since 2021, HPAI H5N1 viruses of the clade 2.3.4.4b severely impacted wild bird species and poultry across Europe ⁹⁻¹¹, southern Africa ¹²⁻¹⁴, North America ¹⁵, and South America where it resulted in die-offs of tens of thousands of seabirds ¹⁶ and marine mammals ^{17,18}."

"it reached South Africa 7". The authors are talking about H5N1 here, but it was an H5N8 (of the goose Guangdong HA lineage) that reached south Africa, Again the language in the introduction could use some cleaning up to specify this is related to the H5HA, not more specifically H5N1.

Response: We thank the Reviewer for pointing out those imprecisions and have modified the first paragraph of the introduction accordingly (see our response to your previous comment). We initially wanted to cite both the H5N8 of 2016-2017 and the H5N1 of 2021 in South Africa, hence the confusion.

"devastating effect on seal" – I believe most early mortalities were sea lions rather than seals (elephant seals and fur seals were later in Argentina) – maybe worth using a collective term for these mammals such as pinnipeds or semi-aquatic mammals.

Response: We agree that most pinniped mortalities on the West coast of South America were sea lions, however, seal is a generic name for true seals and eared seals, hence it shouldn't be a problem to refer to sea lions under this term. As Reviewer #1 pointed out for 'Procellariiformes', readers might not know high taxonomic names, hence we prefer to keep the term 'seals' rather than 'pinnipeds'.

Although some details are given, It might be worth including some more details of the location of these French Austral lands and seas, relative to mainly Antarctica, South Africa and South Georgia (distance and direction in km) in the introduction.

Response: We thank the Reviewer for the suggestion. As suggested, we added the relative distances between the French islands, mainland Antarctica, South Africa and South Georgia, but in the legend of Figure 4, not in the introduction, as the map helps having a good representation of the relative location of the islands. To this end, we have added line 266-268: “In turn, the Crozet and Kerguelen Islands are respectively approximately 2500 and 3800 kilometers from South Africa, 2200 and 2000 kilometers from Antarctica and 5300 and 4000 kilometers from Australia.”

“pressure from global change” – possibly the authors mean climate change, or global warming here specifically?

Response: Global warming is not not the only threat we wanted to refer to. Increased competition or predation by newly introduced invasive species, reduction of food sources, increase of bycatch due to increase in fishing pressure are other major ones. We replaced “from global change” by “from other threats” (line 81) to highlight the multiplicity of those threats.

Results:

Any data on sex ratios, if available, could be good to include. Previous studies have found that some species have infection rates that bias in favour of a particular sex: <https://doi.org/10.1080/00063657.2024.2396563>. For elephant seals in particular this would be of interest as there are significant sex-related differences in feeding strategies, metabolism etc

Response: We unfortunately don't have such data; carcass scavenging makes it difficult to assess the sex with morphological differences, and we did not do any molecular sexing.

Figure 3 – the colours are a little hard to read on panel A and B – could something like shaped also be included to make the figure clearer to read?

Response: We have modified the colour scheme of panel A - and made the corresponding adjustments in our other figures - to improve the readability of the colours (see our responses to previous comments from the other Reviewers and the phylogenetic tree that is part of this response letter). We think that panel B suffered from quality issues and have worked to include a higher quality version of this panel into our updated figure.

'Molecular markers' – Looking at the sequences there is also a second sequence with mixed E627E/K (24P021416_H5N1_D-24-09077_HP_B3.2_CRO_2024-10-20). My feeling is that mixed mammalian adaptations usually implies direct spillover from an avian source (rather than sustained mammal to mammal transmission) as these mutation is arising de novo – It might be interesting to mention this second sequence and maybe investigate the proportion of 627E vs K reads within in.

Response: We thank the Reviewer for the interesting comment. Given that we obtained this sample from a king penguin carcass and not from an elephant seal carcass, we rather think it could result from simultaneous exposure to and infection with two different viruses rather than a de novo mutation that would have occurred within this host. We also want to remain cautious about not overly interpreting this finding, as it pertains to a single sample. Hence, we have added a sentence in the Results section to discuss this sample, line 276-279: “One of our sequences from a southern elephant seal analysed (A/southern_elephant_seal/Crozet/24P021415/2024) also features an E627K mutation in the PB2 protein, and another one from a king penguin (A/king_penguin/Crozet/24P021416/2024) features a mixed infection with E627K/E.”

The serology aspect of this paper is generally good but a bit rudimentary- given the limitations of the environment I suppose this is to be expected. Possibly one or more of the following might help develop this a bit further:

- Testing of samples (particularly ELISA-positive or inconclusive samples) with an additional assay, e.g. haemagglutination inhibition or neutralisation assay
- To what extent (if any) do the authors think cross-reactivity with seal influenza viruses might impact results?
- Why is serology only performed for elephant seals and not avian species- was this a deliberate study choice or a constraint of sampling?

Response: We agree that the serology aspect of the manuscript is limited, but we think that it is bringing some important qualitative insight in the context of the reported emergence and the very limited immunological knowledge currently available on elephant seal populations affected by HP AIV. To answer to the three specific comments:

- We ran two additional serological analyses to confirm the positive samples: the 3.0 multispecies anti-H5 competitive ELISA from Innovative Diagnostic and an indirect anti-H5 ELISA from Innovative Diagnostics which we modified by using an anti-canine secondary antibody. The 3.0 anti-H5 competitive ELISA was validated for mammals. Those additional analyses confirm the already identified positive pups sampled in 2024 on Kerguelen as opposed to negative samples from earlier. We now present those results in the main manuscript text and in Extended Figure 9 (see below).
- We did not expect any particular cross-reactions and we are confident of the specificity of the results given the assays we ran. Moreover, the H5 ELISA results of the years before the outbreak show no positive individuals, hence if there would be any strain of seal LP AIV in circulation, we are confident that it does not cross-react with our assay.
- We have actually collected several hundreds of plasma samples from several avian species over the breeding season, in addition to samples collected over several previous years, but because of logistical constraints and to keep things focused / concentrated, we decided to already include these first seal serological results in this manuscript (we are still analysing the others) as we thought it could provide interesting (and hopeful) insights into the immunization dynamics of the southern elephant seals, and most importantly to encourage more serological surveys, as at this stage very few serological studies of HPAI 2.3.4.4b in wildlife have been published. We agree though that at this stage it is rudimentary, but it stresses two important things: (i) elephant seals had likely not been exposed to H5 avian influenza viruses in the years before 2024, (ii) some pups responded by mounting an humoral immune response.

We have hence added the following elements in the main text to account for those precisions:

- in the discussion (line 365-369): “Our serology results, although small-scale and stemming from the beginning of the outbreak, provide hope that a proportion of exposed southern elephant seals become immune and might survive the infection. However, given the limitations aforementioned, we invite to take those results as a first insight and as a call to conduct further investigation into the immunization dynamics.”
- Extended Data Figure 9 (line 651-658):

Extended Data Fig. 9 | Competitive and indirect ELISA for H5 proteins performed on southern elephant seal plasmas. Top left plot is a reproduction of Figure 5 for comparison purposes. The grey area displays the range for inconclusive H5-seropositivity results, according to the kit manufacturer. We introduced a jitter to displace the data points horizontally to improve their visibility. Below right plot shows the correlation between the v2 and v3 multispecies H5 ELISA. The results confirm that two of the pup samples were seropositive against H5 AIV protein in 2024, with no evidence of exposure to H5 AIV prior to that season.”

Discussion – something absent in the discussion is the absence of the sustained mammalian-adapted B3.2 lineage (the one with PB2 D701N/Q591K, described in detail in Uhart et al). The absence of this virus, despite its presence in the Falklands islands, and also in the mainland Antarctic, might suggest elephant seals are less likely to be the vectors of this virus to the southern Indian/subantarctic islands.

Response: We thank the Reviewer for this suggestion. We have added a sentence in the discussion, line 371-375: “Further, the clade that was dominant in South America among marine mammals was not the one spreading to South Georgia Island, nor to the Indian Ocean, despite the fact that – as in South America – southern elephant seals were strongly affected on South Georgia, the Crozet and the Kerguelen Islands. This may suggest that the seals are unlikely long-distance vectors of the virus.”

We would like to keep this element of discussion short though, as this is in our opinion an argument that was already valid last year when the virus (not the mammalian-adapted lineage) reached South Georgia. The further spread from South Georgia to the Indian Ocean does not necessarily provide additional evidence (i.e., the surprise was that the marine mammal clade did not reach South Georgia in the first place - but now the further spread from South Georgia to the Indian ocean, rather than from South America to the Indian Ocean, makes sense as South Georgia is more connected to the rest of the sub-Antarctic than South America).

Also – although I understand the phylogenetic analysis was done prior to the handful of mainland Antarctic H5 sequences being shared publicly, it would still be good to mention where they would fall in the phylogenetic trees. Including them in some phylogenetics on the authors new sequences shows a few of these sequences fall fairly close to the south Indian ocean sequences, at least for a few of the segments, particularly the sequences from Torgersen Island (A/Brown_Skua/Torgersen_Island/o81-b82/2024|EPI_ISL_19645365, A/Brown_skua/Torgersen_Island/o8182/2024|EPI_ISL_19745586). These sequences could majorly change the interpretation of the phylogenetics, suggesting spread to Ker/Cro from the mainland Antarctic, rather than by the subpolar route.

Response: We thank the Reviewer for this suggestion. As mentioned in our response to Reviewer #2, there are four recently submitted (on April 24th, 2024) genomic sequences from Antarctica in addition to the two from Torgersen Island that the Reviewer is referring to (all from brown skuas). As shown in an earlier figure in this response letter, the inclusion of these sequences into our analysis pipeline did not change the interpretation of the phylogenetics as they were also estimated to have originated from South Georgia through an independent introduction. As such, our conclusion regarding the Kerguelen and Crozet introductions remain the same as in our original manuscript.